# Identification of compound drought and heatwave events on a daily scale and across four seasons

Baoying Shan[1,2], Niko E.C. Verhoest[2], and Bernard De Baets[1]

[1]KERMIT, Department of Data Analysis and Mathematical Modelling, Ghent University, 9000 Ghent, Belgium
[2]Hydro-Climatic Extremes Lab, Ghent University, 9000 Ghent, Belgium

**Correspondence:** Baoying Shan (baoying.shan@ugent.be)

**Abstract.** Compound drought and heatwave (CDHW) events can incur intensified damage to ecosystems, economies, and societies, especially on a warming planet. Although it has been reported that CDHW events in the winter season can also affect insects, birds, and wildfires, the literature generally focuses on the summer season exclusively. Moreover, the coarse temporal resolution of droughts as determined on a monthly scale may hamper a precise identification of the start and/or end dates of
5 CDHW events. Therefore, we propose a method to identify CDHW events on a daily scale that is applicable across the four seasons. More specifically, we use standardized indices calculated on a daily scale to identify four types of compound events in a systematic way. Based on the hypothesis that droughts or heatwaves should be statistically extreme and independent, we remove minor dry or warm spells and merge mutually dependent ones. To demonstrate our method, we make use of 120 years of daily precipitation and temperature observed at Uccle, Brussels-Capital Region, Belgium. Our method yields more precise
start and end dates for droughts and heatwaves than what can be obtained with a classical approach acting on a monthly scale, thus also allowing for a better identification of CDHW events. Consistent with existing literature, we find an increase in the occurrence of the number of days in CDHW events at Uccle, mainly due to the increasing frequency of heatwaves. Our results also reveal a seasonality of CDHW events as droughts and heatwaves are negatively dependent in the winter season at Uccle, while they are positively dependent in the other seasons. Overall, the method proposed in this study is shown to be robust and
shows potential for exploring how year-round CDHW events influence ecosystems.

## 1 Introduction

Compound events are defined as a combination of multiple drivers and/or hazards that could lead to extreme ecological and socioeconomic impacts of larger magnitude than the sum of those of the individual ones (Zscheischler et al., 2018; Masson-Delmotte et al., 2021). Under the challenge of climate change, the extremes, especially those related to temperature, are
20 observed at an increasing pace (Perkins et al., 2012; Sharma and Mujumdar, 2017; Mukherjee and Mishra, 2021; Kruger and Sekele, 2013) and are expected to become even more severe in the future (Byrne, 2021; Wu et al., 2021). It is critical to study heatwaves concurrently with droughts, because of the intensification of negative impacts, such as exacerbating water shortage (Coffel et al., 2019), crop failure, and gross primary production reduction (Ciais et al., 2005; Mishra et al., 2020; Zampieri et al., 2017), wildfire and tree mortality (Libonati et al., 2022; Brando et al., 2014; Reichstein et al., 2013), and so on.

Recently, the increasing frequency of compound drought and heatwave (CDHW) events (or compound hot and dry events), has received attention, both globally (Mukherjee and Mishra, 2021; Zhang et al., 2022) and regionally, e.g. in the United States (Mazdiyasni and AghaKouchak, 2015; Alizadeh et al., 2020), India (Sharma and Mujumdar, 2017), Europe (Manning et al., 2019), China (Kong et al., 2020) and Southeast Brazil (Geirinhas et al., 2021).

The identification of CDHW events lies at the core of temporal trend and frequency analyses. However, the coarse temporal resolution and scale inconsistency of the indices used hamper the reliability of existing studies. Droughts are commonly identified on the basis of monthly magnitude variations in historical climate variables (McKee et al., 1993; Ridder et al., 2020; Salvador et al., 2020), and therefore completely ignore the intramonthly distribution, while heatwaves are usually obtained on a daily scale. For example, the *3-month* Standardized Precipitation Index (SPI) for meteorological droughts and the *daily* temperature index for heatwaves are a popular pair of indices that are used for identifying compound events (see, e.g., Geirinhas et al. (2021)). This coarse scale of the drought index and the mismatch of scales between the drought and heatwave indices could entail some bias on the actual start and/or end dates as well as the severity of droughts, further affecting the identification of CDHW events. Recent studies have upscaled the scale of drought events to a weekly, 5-day, or daily scale (Mukherjee and Mishra, 2021; Wang et al., 2020; Li et al., 2020; Mo and Lettenmaier, 2015; Yuan et al., 2023). For example, Mukherjee and Mishra (2021) identified a drought week when the self-calibrated Palmer Drought Severity Index (Wells et al., 2004) falls below the 10th percentile, thus regarding short dry spells (a week) as drought events; Wang et al. (2020) calculated daily Standardized Precipitation Evapotranspiration Index (SPEI) values and identified a drought event as a period of consecutive days with SPEI below a given threshold (such as -1.0). While these identification methods enhance the temporal resolution, they also introduce a new challenge. They may result in a large number of short dry spells and/or mutually dependent ones. Consequently, the droughts identified might not be extreme or independent. Therefore, additional efforts are needed to address the issues of coarse resolution and scale inconsistency.

Most studies on CDHW events exclusively focus on the summer or extended summer season, mainly because heatwaves that lead to deadly heat stress for human beings and ecosystems are only expected in this period of the year (McKechnie and Wolf, 2010; Ummenhofer and Meehl, 2017; Stillman, 2019; Soroye et al., 2020). While being overlooked in the study of CDHW events, ecologists have studied the (ecological) impact of relatively high temperatures (and droughts) in non-summer seasons. For example, warmer winters and earlier springtime drying of soils and forest fuels (fallen leaves) are probably linked to increasing numbers of large wildfires and total area burned in western US forests (Westerling et al., 2006; Littell et al., 2009; Williams et al., 2010), which could weaken the resistance of surviving trees against bark beetle attack (Raffa et al., 2008). Laboratory experiments conducted by Radchuk et al. (2013) and Oliver et al. (2015), and a long-term countrywide data set analysis by McDermott Long et al. (2017) all found that elevated temperatures during the overwintering period could increase the rates of mortality for UK butterflies, possibly due to the increased incidence of diseases and fungi (Radchuk et al., 2013) or decoupling from photoperiod cues (Wiklund et al., 1996). More studies on the wild passerine in France (Marrot et al., 2017), the Great tit (a songbird) in UK (Cole et al., 2021) and bee colonies across Austria (Switanek et al., 2017) indicate the significant ecological influence of relatively high temperatures. Hence, from an ecological point of view, a tool that is able to identify CDHW events across all seasons is badly needed.

Zscheischler et al. (2020) proposed a typology of compound hazard events focusing on four themes: preconditioned, multivariate, temporally compounding, and spatially compounding. For the multivariate events where multiple hazards occur at the same time (for example, droughts and heatwaves in this study), existing literature generally identifies such compound events as the intersection of the periods of the co-occurring hazards (Zscheischler and Seneviratne, 2017; Ridder et al., 2018; Mukherjee and Mishra, 2021). However, from a more systematic point of view, we can distinguish various ways to determine the start and end dates of such compound events: the mentioned intersection, but also the union, or one of the periods conditioned on the other.

The overall objective of this study is to develop a method to identify CDHW events on a daily scale across four seasons by addressing the following research questions:

(a) Can we consistently define droughts and heatwaves?

(b) How can we define heatwaves across four seasons?

(c) How can we avoid the fact that the use of indices calculated on a daily scale could produce a large number of small-scale events and/or mutually dependent events when using fixed-threshold-based identification approaches?

(d) How should we interpret the term 'compound' and how can we retrieve CDHW events from the droughts and heatwaves themselves?

(e) Do CDHW events in non-summer seasons have similar characteristics as those in the summer season?

(f) Are the temporal and frequency trends observed consistent with existing studies?

This study is structured as follows. In Section 2, we describe the method developed and the data used. We make use of 120 years of daily data of precipitation and temperature observed at the Belgian meteorological institute in Uccle, Brussels-Capital Region, Belgium. In Section 3, we present and discuss the results obtained, followed by some conclusions in Section 4.

## 2 Materials and methods

This section mainly describes the proposed method for the identification of CDHW events, which includes three steps. First, we define droughts and heatwaves and calculate the indices for addressing research questions (a) and (b). Then, we identify drought and heatwave events from daily indices for addressing research question (c). Finally, we identify CDHW events from drought and heatwave events for addressing research question (d).

### 2.1 Data

Daily minimum/maximum temperatures from 1901 to 2020 were acquired from the climatological station of the Royal Meteorological Institute (RMI) of Belgium in Uccle ($50°47'55''$ N, $4°21'29''$ E, 100 m a.s.l.), while daily rainfall was obtained from 10-minute precipitation time series recorded at the same site. These high-quality observations have been used in many studies, see e.g. Verhoest et al. (1997); Ntegeka and Willems (2008); Vandenberghe et al. (2010); Pham et al. (2018).

## 2.2 Daily drought and heatwave indices

In this study, a heatwave is defined in a relative sense, *i.e.*, as a period of excessively high temperatures compared to the expected normal temperatures in this period. As the expected normal temperatures differ for individual calendar days, heatwaves defined in this manner could capture anomalous warm periods during summer and winter. We use the daily SPI (McKee et al., 1993) and SHI (Standardized Heatwave Index) (Raei et al., 2018) as indices to identify droughts and heatwaves, respectively. SPI (resp. SHI) is a standardized index to quantify to what extent precipitation (resp. temperature) deviates from the climatological average. SPI (resp. SHI) describes dry (resp. hot) spells according to the probability of occurrence in a reference period. Given the non-stationarity of temperature records due to global warming, the historical reference period and climate normals for calculating such probabilities must be carefully selected. Accounting for the fact that people and ecosystems adapt to a changing climate, we use the past 30 years as the historical reference period and average the values on every day during this period. This average is then regarded as the expected normal temperature on this day, instead of the average during the longest climatology (the period of record). This 30-year moving window approach has been suggested to deal with the climate non-stationarity bias (Hoylman et al., 2022). To examine the impact of this approach, we apply the Mann–Kendall (MK) test for the daily mean temperature in the period 1901–2020 (the period of record) and in the 30-year moving windows (1901–1930, 1902–1931,...,1991–2020; 91 windows per day) based on the data described in Subsection 2.1. The test results (see Fig. S1 for details) point out that over the period of record, 40.27% of calendar days (147 out of 365 days) show a significant increase, while not a single day shows a significant decrease. However, for the 30-year moving windows, only 5.59% of windows (33215 windows in total, 91 per day) show a significant trend, 4.38% with an increase and 1.21% with a decrease, indicating that this approach better accounts for climate non-stationarity.

After having overcome the non-stationarity problem in the above way, we calculate SHI values for each day as:

$$
\hat{T}_{m,i} =
\begin{cases}
\dfrac{1}{N_{\mathrm{h}}} \displaystyle\sum_{j=i-N_{\mathrm{h}}+1}^{i} T_{m,j} & , \text{if } i \geq N_{\mathrm{h}} \\[3mm]
\dfrac{1}{N_{\mathrm{h}}} \left( \displaystyle\sum_{j=365-N_{\mathrm{h}}+i+1}^{365} T_{m-1,j} + \sum_{j=1}^{i} T_{m,j} \right) & , \text{if } i < N_{\mathrm{h}}
\end{cases}
\tag{1}
$$

$$
\mathrm{SHI}_{m,i} = \Phi^{-1}(F_{m,i}^{\mathrm{h}}(\hat{T}_{m,i}))
\tag{2}
$$

where $i$ and $j$ represent calendar days ($i, j \in \{1, 2, \ldots, 365\}$), and $m \in \{1, 2, \ldots, M\}$ the year considered, with $M$ the number of years of historical observations; $T_{m,i}$ (°C) is the daily mean temperature at day $i$ of year $m$, which is obtained by averaging the daily maximum and minimum temperatures. For leap years, we average the daily mean temperature on the 28th and the 29th February and assign it to the 28th February, and then remove the leap days. $N_{\mathrm{h}}$ (d) is the length of the accumulation period for a heatwave. $\hat{T}_{m,i}$ (°C) is the mean temperature over the accumulation period up to and including day $i$; $F_{m,i}^{\mathrm{h}}$ is the cumulative distribution function (CDF) fitted to $\{\hat{T}_{m,i}, \hat{T}_{m-1,i}, \ldots, \hat{T}_{m-29,i}\}$ if $m \geq 30$, otherwise to $\{\hat{T}_{30,i}, \hat{T}_{29,i}, \ldots, \hat{T}_{1,i}\}$; $\Phi^{-1}$ is the inverse standard normal CDF; $\mathrm{SHI}_{m,i}$ is the estimated SHI value at day $i$ of year $m$.

While not relevant for SHI values, a critical problem in the calculation of SPI values is how to deal with the common zeros in precipitation time series (Mishra and Singh, 2010). Indeed, strictly positive distributions (like the gamma distribution) are

undefined at zero, while for a great deal of positive distributions (like the Weibull distribution), the density function at 0 is always 0, no matter how many zero values are present in the data to which they are fitted. If these distributions were used to fit precipitation time series with many zeros, zero and small rainfall amounts (usually indicating dry conditions) would be not well accounted for, which might affect the drought analysis. To resolve this issue, we split the precipitation values into zero values and positive values and estimate their probabilities separately (Naresh Kumar et al., 2009; Farahmand and AghaKouchak, 2015), as expressed in Eq. (4). In this way, we calculate SPI values for each day as:

$$
\text{SP}_{m,i} = \begin{cases} \displaystyle\sum_{j=i-N_\text{d}+1}^{i} P_{m,j} & , \text{if } i \geq N_\text{d} \\ \displaystyle\sum_{j=365-N_\text{d}+i+1}^{365} P_{m-1,j} + \sum_{j=1}^{i} P_{m,j} & , \text{if } i < N_\text{d} \end{cases} \tag{3}
$$

$$
q_i = \frac{Z_i}{M} \tag{4}
$$

$$
F_i^\text{d}(\text{SP}_{m,i}) = \begin{cases} q_i & , \text{if } \text{SP}_{m,i} = 0 \\ q_i + (1-q_i) \times F_i(\text{SP}_{m,i}) & , \text{if } \text{SP}_{m,i} > 0 \end{cases} \tag{5}
$$

$$
\text{SPI}_{m,i} = \Phi^{-1}(F_i^\text{d}(\text{SP}_{m,i})) \tag{6}
$$

where $P_{m,i}$ (mm) is the daily precipitation at day $i$ of year $m$. For leap years, we add the precipitation on leap days to that on the 28th February, and then remove the leap days. $N_\text{d}$ (d) is the length of the accumulation period for a drought. $\text{SP}_{m,i}$ (mm) is the precipitation over the accumulation period up to and including day $i$; $Z_i$ represents the number of zero $\text{SP}_{m,i}$ over the $M$ years; $F_i$ is the CDF fitted to the positive precipitation values $\{\text{SP}_{m,i} \mid \text{SP}_{m,i} > 0, m = 1, 2, \ldots, M\}$; $F_i^\text{d}$ is the CDF of all precipitation values. The SPI value $\text{SPI}_{m,i}$ at day $i$ of year $m$ is then obtained by applying the inverse $\Phi^{-1}$ of the standard normal CDF.

The choice of the distribution to be fitted is an important source of uncertainty in the SPI values (Mishra and Singh, 2010), given that, depending on the case considered, other commonly used distributions are valid (Laimighofer and Laaha, 2022). In this study, we consider 10 commonly used distributions from which we select the one with the lowest value for the Akaike Information Criterion (AIC) (Akaike, 1974) in the calculation of both the SPI and SHI values. The distributions considered are the normal, exponential, gamma, generalized extreme value (GEV), inverse Gaussian, logistic, log-logistic, log-normal, Burr, and extreme value distributions.

## 2.3 Extreme and independent events

Droughts (resp. heatwaves) are commonly identified by a fixed-threshold approach as periods with index values consecutively below (resp. above) a certain threshold (e.g., -1 for SPI), even though some studies use different thresholds for the onset and ending of an extreme event. In this study, we select one threshold and refer to it as the pre-identification threshold and to the resulting events as dry (resp. warm) spells.

Yet, this fixed-threshold approach, especially when applied to daily indices, may result in a large number of small-scale and mutually dependent spells, causing the identified droughts or heatwaves to be not extreme or not independent. In turn,

this could affect the analysis of temporal trends and frequencies. Procedures for excluding minor spells and merging mutually dependent ones have already been explored in literature (Fleig et al., 2006). However, it is still tricky to discern objectively whether spells are minor or whether two neighboring spells are dependent.

In this study, we develop a method to obtain objective thresholds for excluding minor spells and merging mutually dependent ones. First, two indicators are selected: the *duration* (d) and the *proximity*. The duration expresses how long a spell lasts, and
155 equals the number of days in a dry ($D_d$) or a warm spell ($D_h$), while the proximity describes how close two neighboring spells are (Fig. 1). To define the proximity, we first define the total deficiency. The total deficiency of a time interval $[a, b]$ corresponds to the area enclosed between the SPI (resp. SHI) curve and the pre-identification threshold line. The total drought deficiency ($TD_d$) and total heatwave deficiency ($TD_h$) are calculated as:

$$TD_d = \sum_{i=a}^{b}(SPI_{m,i} - SPI_d) \tag{7}$$

$$TD_h = \sum_{i=a}^{b}(SHI_h - SHI_{m,i}) \tag{8}$$

where $SPI_d$ and $SHI_h$ are the corresponding pre-identification thresholds.

The drought proximity $C_d$ (resp. heatwave proximity $C_h$) is defined as the total drought deficiency $TD_d$ (resp. $TD_h$) of the time window between two neighboring spells. The proximity provides a comprehensive view of the period between two spells, accounting for both duration and how far it is from being dry (resp. warm). For example, a small value of $C_d$ means that the
165 interval is short and/or the SPI values are near the pre-identification threshold. In that case, the neighboring spells are more likely to be mutually dependent, suggesting to merge the neighboring spells and the interval between them into one longer spell.

More importantly, in view of our aim of droughts and heatwaves to be extreme and independent, we assume that their severity should follow a Generalized Extreme Value (GEV) distribution (to guarantee that only extreme-impact events are included)
and their arrivals to occur according to a Poisson process (to guarantee that they occur independently in time). Severity is the characteristic used to quantify droughts (resp. heatwaves), usually calculated by means of drought (resp. heatwave) indices. In this study, the severity of a dry (resp. warm) spell is calculated as the negative of the sum of the SPI values (resp. the sum of the SHI values) in a dry (resp. warm) spell. The above hypotheses could be used as objective criteria for determining appropriate thresholds for removing and merging spells, *i.e.*, thresholds that make the severity of the final identified events follow a GEV
distribution, as well as the inter-arrival times to follow an exponential distribution at the same time.

Concretely, we remove a spell when its duration is shorter than a chosen removal threshold ($D_d < R_d$, resp. $D_h < R_h$). After removing such spells, we check whether any proximity is smaller than a chosen merging threshold ($C_d < M_d$, resp. $C_h < M_h$). If this is the case, the two neighboring spells are considered to belong to one longer spell. Therefore, they are merged, and thus the period between both initial spells is now considered to be part of a longer spell. These processes are illustrated in Fig. 1.
The specific steps to obtain the proper thresholds for removing and merging spells, taking the identification of heatwaves as an example, are explained next:

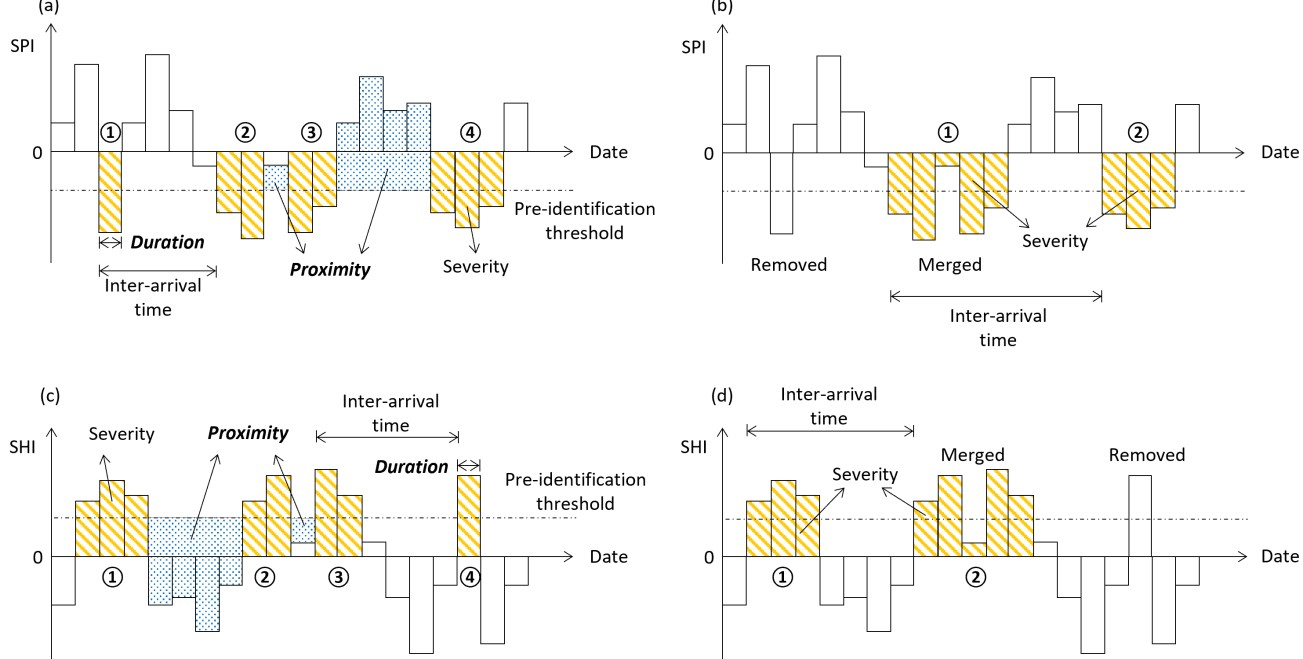

**Figure 1.** Conceptual illustration of the identification of droughts and heatwaves. (a) and (c) show the dry spells and warm spells obtained after applying the pre-identification threshold; (b) and (d) show the final identified droughts and heatwaves after applying the removal and merging procedures. Duration refers to the length of a warm spell or dry spell, while inter-arrival time is the length of the period between the arrivals of two spells. The blue area represents the proximity, while the yellow area represents the severity.

(1) Given an accumulation period length $N_{\mathrm{h}}$ and pre-identification threshold $\mathrm{SHI_h}$, we calculate the daily SHI values using Eqs. (1)–(2).

(2) We label all warm spells potentially belonging to a heatwave (days consecutively above $\mathrm{SHI_h}$).

(3) We consider all possible combinations of removal and merging thresholds $R_{\mathrm{h}}$ and $M_{\mathrm{h}}$, *i.e.*, $R_{\mathrm{h}} \in \{1, \ldots, 5N_{\mathrm{h}}\}$ and $M_{\mathrm{h}} \in \{-\infty, 0, 1, \ldots, 5N_{\mathrm{h}}\}$, where $-\infty$ means no merging is carried out. For each combination the following steps are carried out.

    (3.1) We remove the warm spells with a duration shorter than $R_{\mathrm{h}}$ and relabel the remaining warm spells. Subsequently, we merge neighboring spells if the corresponding proximity $C_{\mathrm{h}}$ is less than $M_{\mathrm{h}}$ and again relabel the warm spells. After removing and merging, the severities (sum of SHI values during a warm spell) and inter-arrival times (d) for this combination are calculated as shown in Fig. 1.

    (3.2) We test whether the inter-arrival times follow an exponential distribution by means of a Kolmogorov–Smirnov test (KS test) (Massey, 1951) to meet the hypothesis of heatwaves being independent. The KS test is chosen because it

does not assume any particular underlying distribution. If the KS test rejects the null hypothesis, we proceed to the next combination; otherwise, we go to step (3.3).

(3.3) We test whether the GEV distribution fits the severities to meet the hypothesis that heatwaves should be extreme. If the KS test rejects the null hypothesis, we proceed to the next combination; otherwise, we go to step (3.4).

(3.4) We check whether other commonly used distributions also fit the severities by means of the KS test. The distributions considered are the same as those used in the calculation of the SPI and SHI values. If the KS test rejects all other distributions, we take a note for this combination that the GEV distribution is the only fit, and proceed to the next combination; otherwise, we go to step (3.5);

(3.5) We compute the AIC and Root Mean Square Deviation (RMSD) values for all distributions that pass the KS test. We record whether the GEV distribution has the lowest AIC or the lowest RMSD.

(4) Among all combinations for which the GEV distribution is the only fit or results in both the lowest AIC and the lowest RMSD values, we select the one leading to the largest number of events in order to increase the probability of having more compound events and benefit the further analyses. If there is no such combination, we consider the combinations resulting in the lowest AIC only. If no combination at all has been identified, then the accumulation period length and pre-identification threshold considered are discarded.

(5) If a combination of thresholds has been identified, then the warm spells obtained after applying the removal and merging procedures are finally labeled as heatwaves, meeting the hypothesis that they are extreme and independent.

The identification of drought events follows a similar procedure.

## 2.4 Identification of compound events

After having identified droughts and heatwaves, we proceed with the identification of CDHW events. In literature, a CDHW event is generally defined as the period of overlap, *i.e.*, the intersection, of a drought and a heatwave (Fig. 2(d)). However, this is not the only way in which a compound event could be defined. In general, we can distinguish at least four different ways to define a compound event. These compound events consist of:

(i) the days belonging to both the drought and the heatwave, *i.e.*, the intersection (d-and-h). Note that a drought might overlap with multiple heatwaves, thus resulting in multiple compound events; the same applies to a heatwave;

(ii) the days belonging to the drought or the heatwave, *i.e.*, the union (d-or-h), provided the events overlap. Note that a drought might overlap with multiple heatwaves, thus resulting in multiple overlapping compound events, that need to be merged subsequently;

(iii) the days belonging to the drought, denoted as d-cond-h, provided the events overlap. Note that a drought overlap with multiple heatwaves. Thus, in the d-cond-h case, only those droughts are preserved that overlap with at least one heatwave;

(iv) the days belonging to the heatwave, denoted as h-cond-d, provided the events overlap. Note that a heatwave might overlap with multiple droughts. A heatwave that does not overlap with at least one heatwave is not considered in the h-cond-d case.

Figure 2 displays a conceptual diagram of the above four different types of CDHW events. A CDHW event is characterized by three variables: the duration, the marginal drought severity, and the marginal heatwave severity. The marginal drought severity is computed as the negative of the sum of the SPI values within the CDHW event, while the marginal heatwave severity is computed as the sum of the SHI values within the CDHW event.

It could be of relevance to ecologists to investigate whether compound events have a higher impact than droughts or heatwaves only. For certain species, it is known that heatwaves have an impact (Pipoly et al., 2021), yet it might be interesting to investigate the possible intensified or weakened impacts of heatwaves during drought conditions (*i.e.*, h-cond-d). Similarly, droughts occurring during a heatwave may have an increased impact (*i.e.*, d-cond-h). For species that suffer from both droughts and heatwaves, a compound event during which a drought and/or a heatwave occur (*i.e.*, d-or-h) may have a stronger impact than when droughts or heatwaves occur independently of each other.

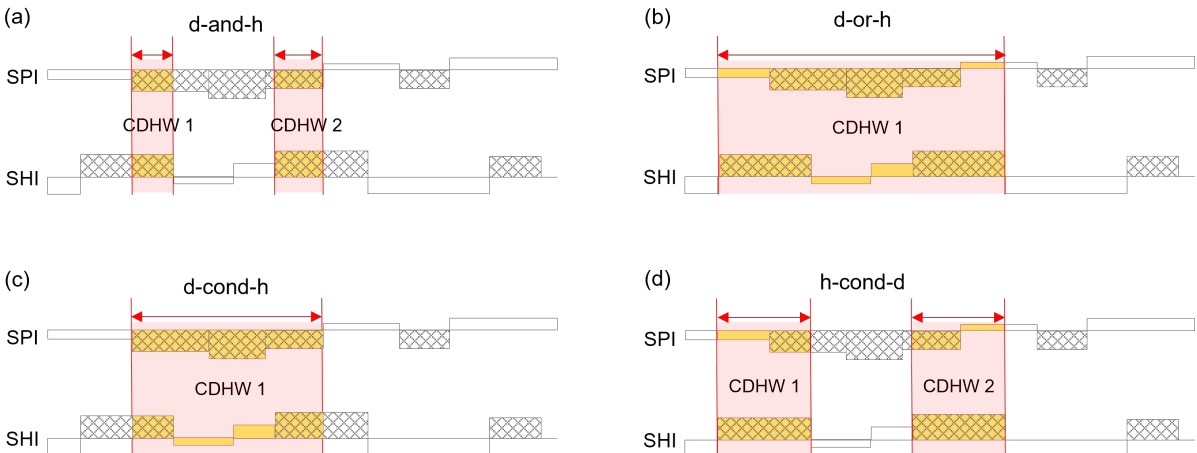

**Figure 2.** Conceptual diagram of four types of CDHW events: intersection (d-and-h), union (d-or-h), drought conditioned on heatwave (d-cond-h), and heatwave conditioned on drought (h-cond-d). Grey shadows represent the days in droughts or heatwaves; Arrows indicate the duration of CDHW events.

## 2.5 Probabilities of occurrence

The probabilities of occurrence of a day being in a drought ($P_{\text{d}}$), a heatwave ($P_{\text{h}}$), or a CDHW event (*i.e.*, $P_{\text{d-and-h}}$, $P_{\text{d-or-h}}$, $P_{\text{d-cond-h}}$, and $P_{\text{h-cond-d}}$) are estimated empirically as the fractions of days belonging to the corresponding events within the study period. By comparing $P_{\text{d-and-h}}$ with the probability of having a compound event assuming that droughts and heatwaves occur independently of each other, *i.e.*, by comparing $P_{\text{d-and-h}}$ with $P_{\text{d}}P_{\text{h}}$, we can examine whether there is a positive dependence

($P_{\text{d-and-h}} > P_{\text{d}}P_{\text{h}}$ ), negative dependence ($P_{\text{d-and-h}} < P_{\text{d}}P_{\text{h}}$), or no dependence at all ($P_{\text{d-and-h}} \approx P_{\text{d}}P_{\text{h}}$) between droughts and heatwaves (Ridder et al., 2020). Further, we calculate the likelihood multiplication factor (LMF) (Ridder et al., 2020), as $\text{LMF} = \frac{P_{\text{d-and-h}}}{P_{\text{d}} \times P_{\text{h}}}$. In the case of independence, LMF approximately equals 1; if droughts and heatwaves are positively dependent,

LMF exceeds 1 and increases with the strength of dependence.

## 3  Results and discussion

### 3.1  SPI and SHI

We calculated the SPI and SHI for various accumulation period lengths ($N_{\text{d}}$ = 15, 30, 45, 60, 90 d; $N_{\text{h}}$ = 3, 5, 7, 10, 15 d). The distributions in the SPI and SHI are selected based on the lowest AIC values. The frequency of the selected distribution is

250 shown in Figure 3. The Gamma distribution is the most frequently selected one in the calculation of SPI, followed by the GEV, Burr, and Normal distributions. For SHI, there is a wider range of distributions, including the Normal, Gamma, GEV, Inverse Gaussian, EV, Logistic, Loglogistic, and Burr distributions.

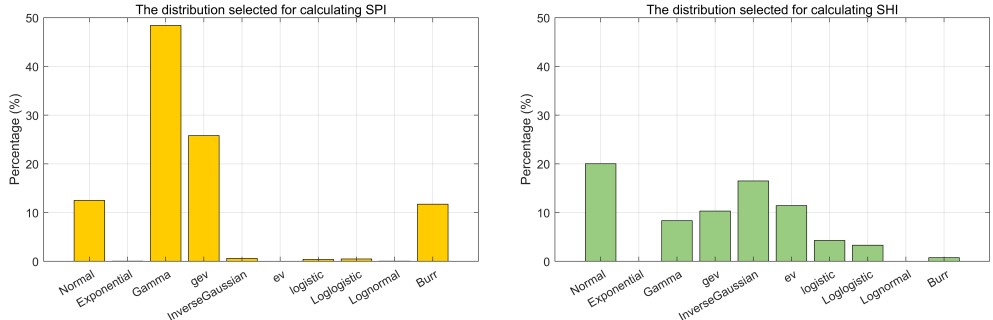

**Figure 3.** The frequency of the distribution selected with the lowest AIC value in the calculation of SPI and SHI

The choice of distribution can significantly influence the SPI and SHI values, especially in the tails of the distribution, which are particularly relevant for extreme events. For example, we conducted a comparison between values calculated using fixed

distributions (commonly chosen: Gamma and GEV for SPI, Normal and GEV for SHI) and the values in our study, depicted in Fig. S2. We observed several problematic SHI values when using the GEV distribution, as indicated by the grey circle. Existing studies, like by Mishra and Singh (2010), Laimighofer and Laaha (2022), also highlighted the impact of the choice of the distribution to be fitted. We suggest choosing the proper distribution according to the goodness-of-fit criterion.

### 3.2  Removal and merging thresholds

We identify the droughts and heatwaves for various accumulation period lengths ($N_{\text{d}} = 15, 30, 45, 60, 90$ d; $N_{\text{h}} = 3, 5, 7, 10, 15$ d) and pre-identification thresholds ($\text{SPI}_{\text{d}} = $ -0.5, -1, -1.3; $\text{SHI}_{\text{h}} = 0.5, 1, 1.3$). Table 1 shows the resulting removal and merging thresholds, the corresponding average number of days per year in an event, and the average number of events per year. We

**Table 1.** Removal thresholds, $R_d$, merging thresholds, $M_d$, average number of days per year in an event, and average number of events per year, for droughts (a) and heatwaves (b).

| (a) | $R_\mathrm{d}$ | | | $M_\mathrm{d}$ | | | Number of days per yr | | | Number of events per yr | | |
|---|---|---|---|---|---|---|---|---|---|---|---|---|
| $\mathrm{SPI_d}$ | -0.5 | -1 | -1.3 | -0.5 | -1 | -1.3 | -0.5 | -1 | -1.3 | -0.5 | -1 | -1.3 |
| $N_\mathrm{d} = 15$ | 23 | 14 | 10 | 1 | 15 | 11 | 37.8 | 28.3 | 21.2 | 1.12 | 1.13 | 1.18 |
| $N_\mathrm{d} = 30$ | 42 | 22 | 20 | 8 | 46 | 1 | 45.7 | 33.6 | 17.9 | 0.65 | 0.68 | 0.55 |
| $N_\mathrm{d} = 45$ | 44 | 31 | 25 | 1 | $-\infty$ | 1 | 59.8 | 30.8 | 18.7 | 0.78 | 0.60 | 0.43 |
| $N_\mathrm{d} = 60$ | 58 | 45 | 24 | 0 | $-\infty$ | 51 | 62.1 | 28.9 | 24.9 | 0.58 | 0.43 | 0.39 |
| $N_\mathrm{d} = 90$ | 81 | 59 | 32 | 1 | 9 | 17 | 67.2 | 30.2 | 23.6 | 0.43 | 0.25 | 0.28 |

| (b) | $R_\mathrm{h}$ | | | $M_\mathrm{h}$ | | | Number of days per yr | | | Number of events per yr | | |
|---|---|---|---|---|---|---|---|---|---|---|---|---|
| $\mathrm{SHI_h}$ | 0.5 | 1 | 1.3 | 0.5 | 1 | 1.3 | 0.5 | 1 | 1.3 | 0.5 | 1 | 1.3 |
| $N_\mathrm{h} = 3$ | 13 | 9 | 6 | 0 | 0 | 3 | 34.9 | 19.2 | 16.4 | 1.83 | 1.56 | 1.88 |
| $N_\mathrm{h} = 5$ | 16 | 10 | 8 | 0 | $-\infty$ | $-\infty$ | 40.6 | 25.5 | 16.4 | 1.73 | 1.83 | 1.47 |
| $N_\mathrm{h} = 7$ | 27 | 11 | 10 | 4 | $-\infty$ | 6 | 23.5 | 29.9 | 16.5 | 0.49 | 1.90 | 1.09 |
| $N_\mathrm{h} = 10$ | 23 | 13 | 12 | $-\infty$ | $-\infty$ | $-\infty$ | 41.1 | 33.6 | 18.6 | 1.23 | 1.76 | 1.08 |
| $N_\mathrm{h} = 15$ | 29 | 17 | 18 | 2 | $-\infty$ | 11 | 49.1 | 33.6 | 15.4 | 1.03 | 1.30 | 0.58 |

consider, for example, the case $N_\mathrm{d} = 15$ d and $\mathrm{SPI_d} = -1$. In this case, the combination $R_\mathrm{d} = 14$ d and $M_\mathrm{d} = 15$, results in 28.3 drought days per year and 1.13 drought events per year on average. The values $R_\mathrm{d} = 14$ d and $M_\mathrm{d} = 15$ indicate that all dry spells shorter than 14 d are removed, and that all neighboring spells with $C_\mathrm{d} \leq 15$ are merged, respectively, to meet the hypothesis that drought events are extreme and independent.

The proposed removal and merging method shows the desired behavior. As could be expected, more extended accumulation periods (*i.e.*, larger $N_\mathrm{d}$ or $N_\mathrm{h}$ values) cause the SPI and SHI values to evolve more gradually over time, resulting in dry or warm spells being longer. Besides, larger pre-identification thresholds for droughts (*i.e.*, larger $\mathrm{SPI_d}$) or smaller pre-identification thresholds for heatwaves (*i.e.*, smaller $\mathrm{SHI_h}$) tend to lead to the inclusion of more days in dry or warm spells, hence also resulting in their duration being longer. We can expect the removal threshold needed to exclude minor spells to increase with larger $N_\mathrm{d}$ or $N_\mathrm{h}$ values as well as with larger $\mathrm{SPI_d}$ or smaller $\mathrm{SHI_h}$ values.

As shown in Table 1, for example, if $\mathrm{SPI_d} = -1$, then $R_\mathrm{d}$ takes as values 14, 22, 31, 45, 59 d when $N_\mathrm{d}$ ranges over 15, 30, 45, 60, 90 d. If $N_\mathrm{d} = 15$, then dry spells shorter than 10, 14, and 23 d are removed when $\mathrm{SPI_d}$ ranges over -1.3, -1, and -0.5. In the same way, if $\mathrm{SHI_h} = 1$, then $R_\mathrm{h}$ takes as values 9, 10, 11, 13, 17 d when $N_\mathrm{h}$ ranges over 3, 5, 7, 10, 15 d. In case $N_\mathrm{h} = 3$, then warm spells shorter than 6, 9, and 13 d are removed when $\mathrm{SHI_h}$ ranges over 1.3, 1, and 0.5. The same behavior is also observed for other accumulation period lengths and other pre-identification thresholds. In summary, the way how $R_\mathrm{d}$ and $R_\mathrm{h}$ change with varying accumulation period lengths and pre-identification thresholds in this study is consistent with expectations.

In contrast to the removal thresholds, no clear patterns can be distinguished for the merging thresholds when varying the accumulation period lengths or pre-identification thresholds in Table 1. Interestingly, the merging thresholds influence the number of events. What stands out in Table 1 is the case with $SHI_h = 1.3$ and $N_h = 15$ d. It then holds that $M_h = 11$, leading to 0.58 heatwaves per year on average, which is considerably less than in other cases with the same accumulation period length or pre-identification threshold. This can be explained by the fact that a larger merging threshold results in neighboring spells being combined, ultimately leading to fewer events.

Another interesting observation in Table 1 is that no merging is carried out in some cases ($M_d = -\infty$ or $M_h = -\infty$). That is because, in the proposed method, the removal procedure is carried out first and, subsequently, the merging procedure. After the minor spells are removed, the inter-arrival time between spells correspondingly becomes larger, and the neighboring spells are less likely to be mutually dependent. So, it could happen that after removing minor spells, the inter-arrival time already follows an exponential distribution. Thus, there is no more need to carry out the merging procedure.

We also apply the fixed-threshold method to see what would happen if we did not carry out the removal and merging procedures. The number of days and events is strongly reduced when using our method compared to the fixed-threshold one, as only extreme and independent events are selected. We find an average 28.3–33.6 d/yr and 17.9–24.9 d/yr in droughts for $SPI_d = -1$ and $SPI_d = -1.3$, respectively. Without the removal and merging procedure, there are 56.8–60.6 d/yr and 35.0–38.4 d/yr for $SPI_d = -1$ and $SPI_d = -1.3$, respectively (see Table S1 for details). Similarly, there are 19.2–33.6 d/yr in heatwaves when $SHI_h = 1$, while there are 68.2–68.8 d/yr in heatwaves before removing and merging. This way, only about half of the days are left in extreme events. Moreover, 35.5%–50.7% of dry spells last less than five days, and 5.6%–22.3% of warm spells only last one day in all cases (Fig. S3 in appendix) if the fixed-threshold method is applied. We doubt that such very short spells influence society or ecosystems, and thus suggest to exclude them from the analysis of climate extremes.

### 3.3 Identification results for CDHW events

Different accumulation period lengths and/or pre-identification thresholds for droughts and heatwaves could result in different CDHW events. We consider, for instance, the CDHW events corresponding to $N_d = 15$ d, $SPI_d = -1$ and $N_h = 3$ d and $SHI_h = 1$. The corresponding removal and merging thresholds are given by $R_d = 14$ d, $M_d = 15$ and $R_h = 9$ d, $M_h = 0$.

Next, we discuss the results for the four different types of compound events. In Uccle, in the period 1901–2020 (the period of record), in total 33 CDHW events of types d-and-h and h-cond-d are identified, and 29 CDHW events of types d-cond-h and d-or-h. Hence, their frequency of occurrence ranges from once every 3.6 years to once every 4.1 years (3.6 to 4.1 year return period). Furthermore, the corresponding numbers of days in these CDHW events are given by 2.9 (type d-and-h), 9.1 (type d-or-h), 8.3 (type d-cond-h), and 3.7 (type h-cond-d) d/yr on average. The low numbers of events and numbers of days illustrate that the identification of CDHW events only considers rare events.

Given that we work on a daily scale, a more precise identification of the start and end dates of drought events is possible, prompting a more precise identification of CDHW events. To demonstrate this, we take the year 1976 as an example, in which extraordinary droughts and heatwaves occurred in Belgium, leading to several forest fires and even requiring restrictions on the use of potable water (KMI, 2017). Our method identifies a heatwave spanning from June 23 to July 10, lasting more than

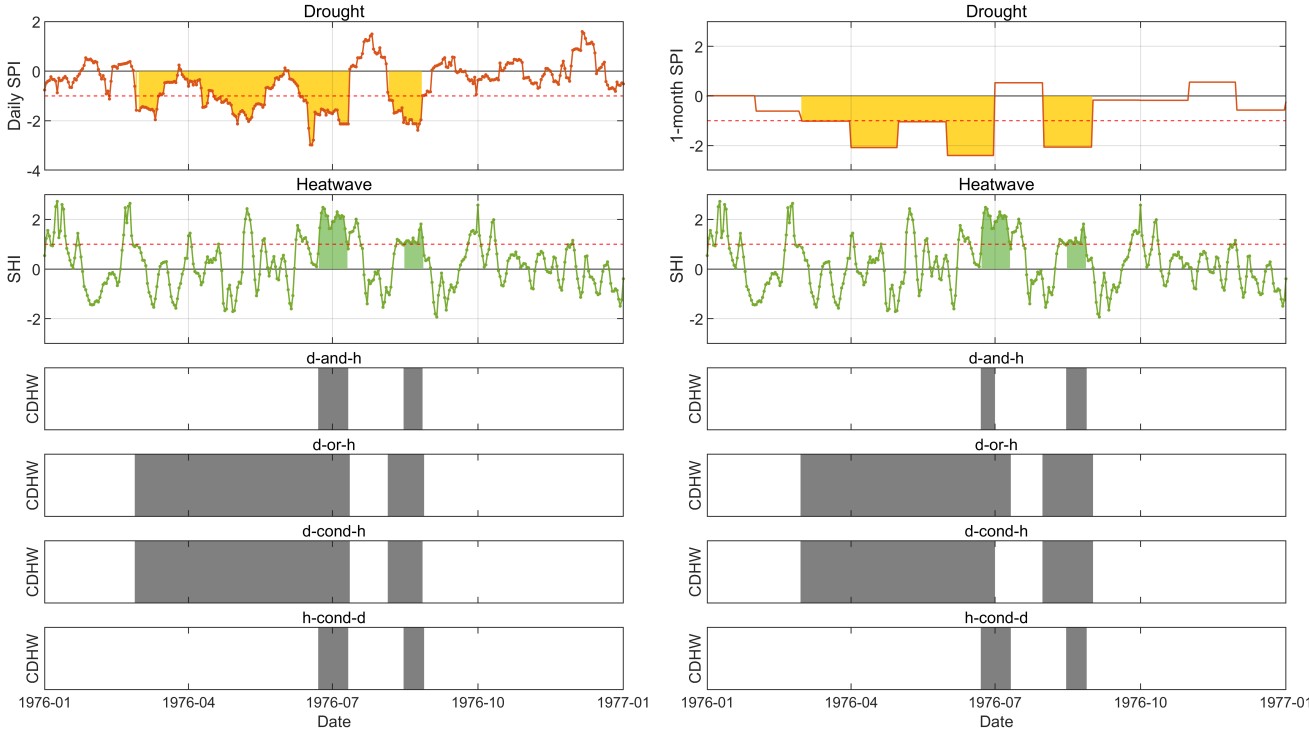

**Figure 4.** Identification results in 1976 for droughts, heatwaves, and CDHW events (of types d-and-h, d-or-h, d-cond-h, and h-cond-d) based on the daily SPI (left) and 1-month SPI (right). Grey areas indicate CDHW events. Identification results in other years are shown in Fig. S4.

half a month, which is exceptionally long for a heatwave in Belgium (shown on the left of Fig. 4). At the same time, there was a severe drought event lasting more than four months, starting on February 28 and ending on July 11 (due to an intense precipitation event (31.7 mm daily rainfall) on July 12). Moreover, a CDHW event of types d-or-h and d-cond-h is identified starting on February 28 and ending on July 11, while a CDHW event of the other two types is identified from June 23 to July 10. Applying a 1-month SPI (shown on the right of Fig. 4), the drought is identified as ending on June 30. Consequently, the corresponding compound event of type d-and-h then starts on June 23 and ends on June 30, lasting 8 d only, in contrast to 18 d in case the daily SPI is used. Comparing these two results reveals that the use of the monthly index could significantly bias the estimation of the duration of CDHW events. The proposed method offers a more precise identification as it is able to capture the start and end dates of droughts and CDHW events intramonthly thanks to the use of daily indices.

Moreover, the method proposed also allows to identify heatwaves and CDHW events in the winter season because the SPI and SHI treat each day in the same way by evaluating extreme anomalies. Figure 5 shows an example of the winter of 2015 when Uccle had an exceptionally mild winter. An average temperature of 6.6 °C was recorded from November 1 until the end of December, while the average temperature in this period over the past 30 years was 3.8 °C. A CDHW event of type d-and-h is

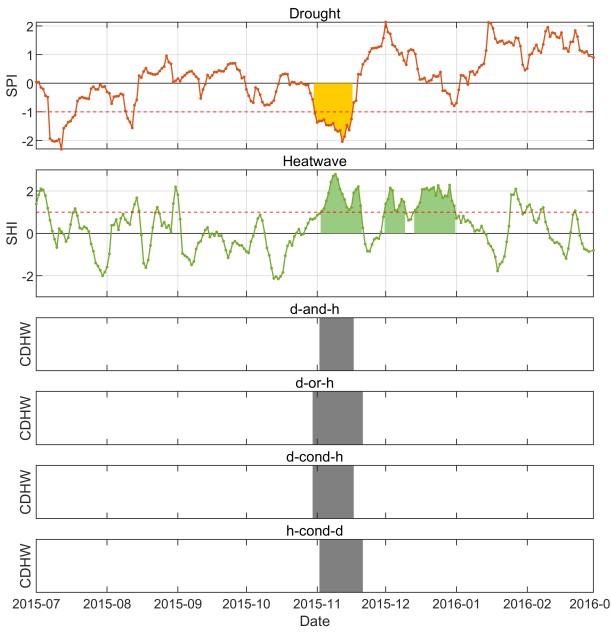

**Figure 5.** Identification results for a winter event in 2015. The orange color is used for droughts, the green color for heatwaves.

identified from November 3 to November 17. Given that CDHW events in non-summer months are also of interest to ecologists (Williams et al., 2010), our method could be of high interest to them.

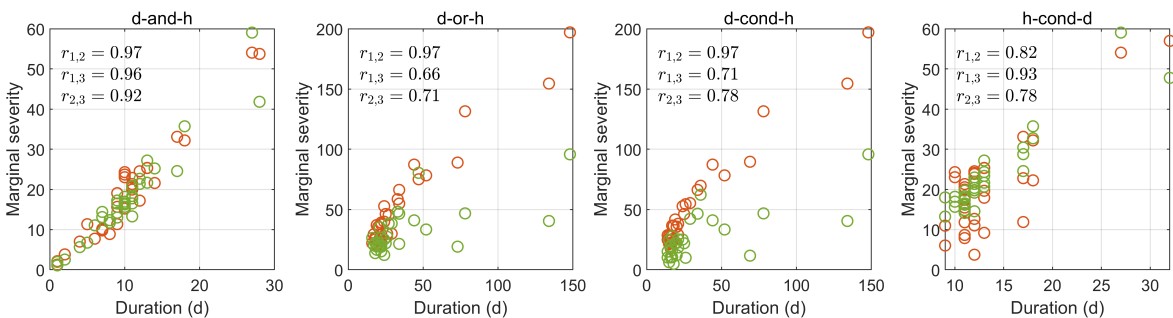

**Figure 6.** Scatter plots for the duration and marginal severity of CDHW events of four types. $r$ stands for the Pearson correlation coefficient, with the subscripts 1, 2, and 3 representing the duration, marginal drought severity, and marginal heatwave severity, respectively. The orange color is used for droughts, the green color for heatwaves.

CDHW events can be described by the following numerical characteristics: the duration, the marginal drought severity, and the marginal heatwave severity. The pairwise relationships between these three characteristics are explored (see Fig. 6): all show positive correlations for the four types of CDHW events. Especially for type d-and-h, the three characteristics are highly

330

correlated, with Pearson correlation coefficients larger than 0.9. Interestingly, CDHW events of type d-or-h and d-cond-h tend to have a longer duration (with a maximum duration of 148 d), while the duration of CDHW events of type h-cond-d and d-and-h is much shorter (with a maximum duration of 32 d and 28 d, respectively), because the length of heatwaves limits the duration of these events.

## 3.4 Temporal trends

In this subsection, we specifically analyze the CDHW events corresponding to $N_d = 15$ d, $SPI_d = -1$ and $N_h = 3$ d, $SHI_h = 1$. However, the trends observed are in line with those for other accumulation period lengths and pre-identification thresholds (see Table S2).

We can observe that the number of CDHW events and the number of days in CDHW events in Uccle are increasing over the period 1901–2020 (see Fig. 7). In particular, the number of CDHW events shows a large increase from 0.167 events/yr in the period 1901–1960 to 0.317 events/yr in the period 1961–2020 for types d-or-h and d-cond-h (or from 0.217 to 0.333 events/yr for types d-and-h and h-cond-d), nearly the double. Also the numbers of days in the four types of CDHW events are higher in the period 1960–2020 compared to the period 1901–1960, such as a rise from 7.78 d/yr to 10.4 d/yr for CDHW events of type d-or-h and from 2.23 d/yr to 3.50 d/yr for CDHW events of type d-and-h. We further applied the MK test to check whether there is a statistically significant trend for the number of days per year in CDHW events. Despite the positive trends for the CDHW events of all four types considered, they do not appear to be significant at the 95% confidence level.

The increasing frequency of heatwaves contributes to the increasing frequency of CDHW events. It should be underlined that heatwaves are obtained through comparison with the expected normal temperatures during the past 30 years instead of the period of record. Nevertheless, there has been a substantial rise in the number of days in heatwaves per year and the number of heatwaves, increasing from 15.6 d/yr and 76 events in the period 1901–1960 to 22.8 d/yr and 111 events in the period 1961–2020. In addition, the MK test for the number of days in heatwaves per year confirms a significant increasing trend at the 95% confidence level. For droughts, there is a slight yet non-significant increase in the number of days in droughts per year, despite the rise in the average number of days in droughts per year and the number of droughts from 27.3 d/yr and 63 events in the period 1901–1960 to 29.3 d/yr and 72 droughts in the period 1961–2020. Thus, the increases for the CDHW events are mainly due to the corresponding increases for heatwaves. This is consistent with the study of Manning et al. (2019) analyzing dry and hot events over Europe in the summer season.

## 3.5 Seasonality

Next, we examine the seasonality of CDHW events (Fig. 8). Their probabilities of occurrence show monthly characteristics. For instance, CDHW events in April, August, and October all have relatively high probabilities of occurrence compared to the other months, regardless of the type of CDHW event considered (d-and-h, d-or-h, d-cond-h, or h-cond-d). Contrarily, the probability of experiencing a compound event is low in December, January and February. For the month of January in particular, no CDHW events have been identified in the period 1901–2020. The limited number of events may preclude the reliable estimation of the monthly probability of occurrence. Alternatively, we can also calculate the probability of occurrence

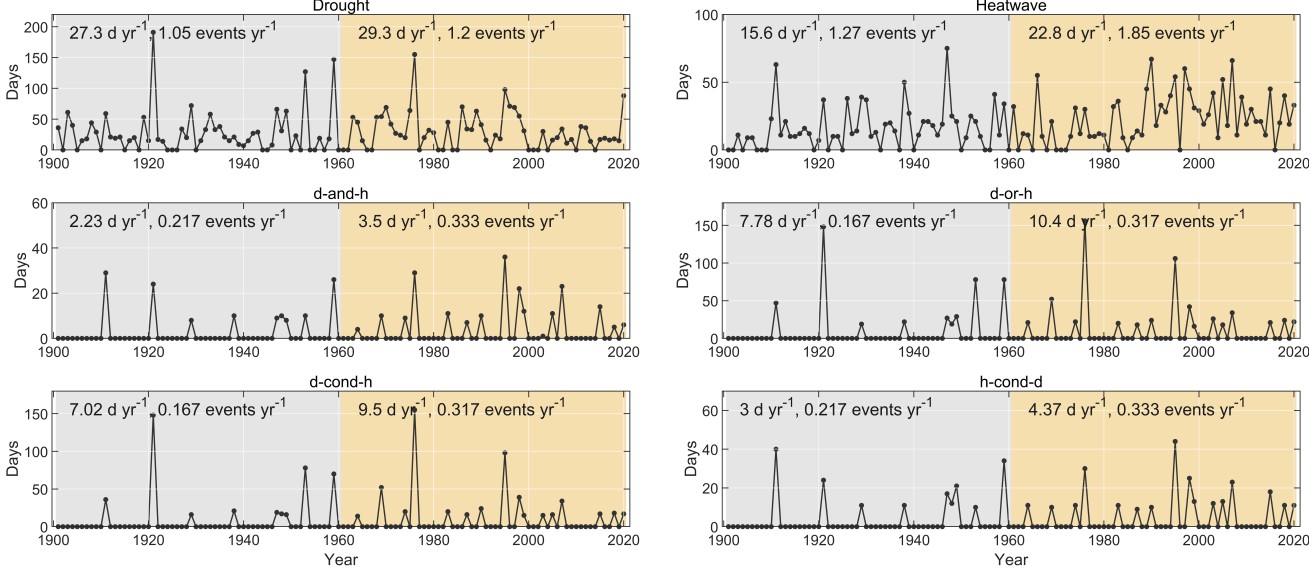

**Figure 7.** Temporal trends of droughts, heatwaves, and CDHW events of four types. Two periods are considered: 1901–1960 (gray) and 1961–2020 (yellow). The superimposed text indicates the number of days in these events per year and the number of events per year averaged over the corresponding period.

at a seasonal scale. The results show that the probability of occurrence of CDHW events in winter is smaller than in the other three seasons, but this difference is not apparent for droughts or heatwaves.

We further compare the probability of occurrence of CDHW events of type d-and-h ($P_{\text{d-and-h}}$) and the probability of having a compound event assuming that droughts and heatwaves are independent ($P_{\text{d}}P_{\text{h}}$). As can be seen from Fig. 8, $P_{\text{d-and-h}}$ is smaller than $P_{\text{d}}P_{\text{h}}$ in winter, while it is greater in spring, summer, and autumn. This reveals that droughts and heatwaves in Uccle tend to coincide in spring, summer, and autumn, while CDHW events are less likely in winter.

Atmospheric and land-atmosphere interactions could explain the above seasonal differences. It has been widely reported and interpreted in literature (see, e.g., Geirinhas et al. (2021); Schumacher et al. (2019); Miralles et al. (2014)) that droughts and heatwaves in summer are positively dependent. First, this phenomenon has been interpreted in the context of soil moisture–atmosphere coupling, with soil moisture deficits related to reduced precipitation leading to enhanced surface sensible heating and higher surface temperatures (Berg et al., 2015). Second, heatwaves could reduce the total energy transfer to the atmosphere, possibly decreasing convective precipitation (Zaitchik et al., 2006). This, in turn, leads to a soil-precipitation feedback loop that tends to extend or intensify drought conditions (Mazdiyasni and AghaKouchak, 2015). Third, the synoptic-scale weather systems favorable for extreme heat are also unfavorable for rain, which increases the probability of co-occurrence (Berg et al., 2015). Precipitation in Uccle is characterized by cyclonal and convective rainfall in the summer months (Verhoest et al., 1997). The above three mechanisms are possible explanations for the simultaneous occurrence of CDHW events in the non-winter seasons at Uccle. In winter, negative correlations between droughts and heatwaves dominate in Uccle as warm moist advection

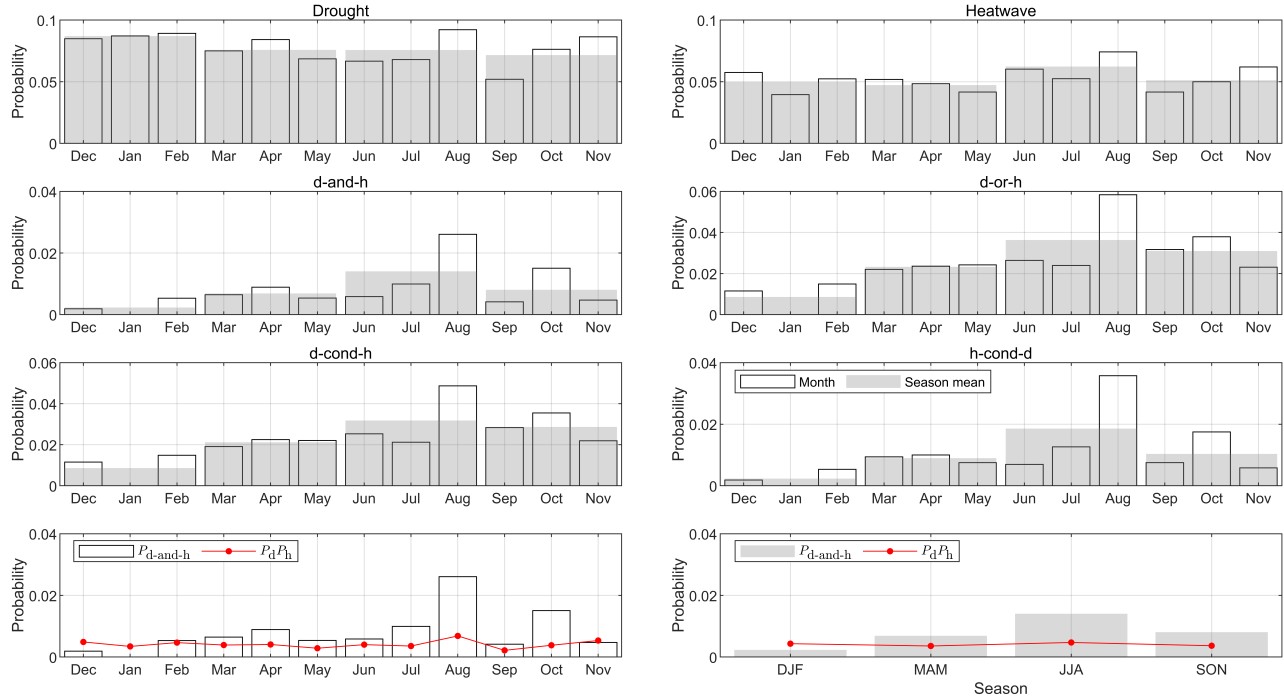

**Figure 8.** Seasonal characteristics. The first three rows show the probabilities of occurrence of a day being in a drought, a heatwave, or a CDHW event of types d-and-h, d-or-h, d-cond-h, and h-cond-d, in each month and each season. The bottom two figures compare $P_{\text{d-and-h}}$ to $P_{\text{d}}P_{\text{h}}$ in each month and each season, respectively.

in extratropical cyclones favors precipitation (Trenberth and Shea, 2005). Uccle is located in the westerlies area with the prevailing wind from west to east. The westerlies are strongest in the winter hemisphere; the pressure is lower over the poles during this time. The warm winter west wind in Uccle favors precipitation causing the CDHW events to be less frequent. Furthermore, due to the radiative influence of clouds, cloudy (often wet) days in winter are generally warmer while sunny (dry) days are colder. Finally, CDHW events develop less likely in winter as soil-precipitation feedbacks play no role as the low atmospheric demand can easily be met by the typically wet soils (Seneviratne et al., 2010; Schumacher et al., 2022).

### 3.6 A small validation experiment

As is well known, it is challenging to validate the identification of drought, heatwave, and CDHW events due to the absence of a universally accepted definition, and the difficulty in measuring the exact start and end dates. We designed an experiment to validate the identification indirectly and compare our proposed method with the commonly used method in other studies. The experiment is based on the well-established physical mechanism that a strong positive land-atmospheric feedback exists between droughts and heatwaves during summer. So, droughts and heatwaves are highly dependent in summer, thus generating a higher probability of CDHW events than when droughts and heatwaves are independent.

The identification method usually comprises two (or three) steps: pre-identification and removal (with merging in some studies). These steps categorize spells into two distinct groups:

A: Final identified drought and heatwave events;

B: Removed dry and warm spells.

Furthermore, when coupled with known physical mechanism, if the identification method is effective, we can expect that during summer, the removed spells in group B would exhibit less dependence than the events in group A. Employing the LMF to quantify the strength of the dependence, we expect the following behavior during summer:

**(1) LMF$_A$ should be greater than 1;**

**(2) LMF$_A$ should be greater than LMF$_B$.**

We consider all possible combinations of different accumulation period lengths and/or pre-identification thresholds for droughts and heatwaves: $N_d = 15, 30, 45, 60, 90$ d, $N_h = 3, 5, 7, 10, 15$ d, $SPI_d = $ -0.5, -1, -1.3, $SHI_h = 0.5, 1, 1.3$, resulting in $5 \times 5 \times 3 \times 3 = 225$ scenarios in total. For each scenario, we calculated LMF$_A$, LMF$_B$, and LMF$_A -$ LMF$_B$ in June, July and August for our proposed removal and merging method, and also for the commonly used method in existing studies by employing a subjective and fixed minimum duration for excluding minor spells. For example, many studies retain drought events only if they persist for more than 30 days to focus on significant events (Brunner and Stahl, 2023; Xu et al., 2023; Christian et al., 2021). Similarly, heatwave events are commonly identified as lasting at least three consecutive days (Ridder et al., 2020; Yin et al., 2023).

Validation results (Figs. S5 and S6) show that LMF$_A$ consistently exceeds 1 for both two methods, which follows the expected behavior (1). However, a notable distinction arises in the expected behavior (2). In all 225 scenarios examined, LMFA $-$ LMFB consistently takes positive values, indicating our method's robustness. In contrast, for the fixed removal method, the condition LMFA $-$ LMFB $> 0$ is not fulfilled in 36 out of 225 scenarios,*i.e.*, in 16% of cases, suggesting that the removed dry and warm spell events had a higher degree of dependence than the final identified events.

Overall, applying a fixed removal threshold method for various accumulation periods and pre-identification thresholds of droughts and heatwaves introduces the potential risk of generating unreasonable results. In contrast, our removal and merging method effectively addresses this challenge during validation.

## 4 Conclusion

We proposed a method for identifying droughts, heatwaves, and compound events. The identification on a daily scale systematically and objectively removed minor spells and merged mutually dependent ones. The analysis conducted at Uccle demonstrates the effectiveness of the proposed method in four ways. First, the values of removal thresholds exhibit the desired behavior, adapting effectively to varying accumulation periods and pre-identification thresholds. Second, the frequency

of occurrence of heatwaves and CDHW events has increased in the period 1961–2020 compared to 1901–1960, and the increasing temperatures contribute to the increase in CDHW events, which aligns with the current literature (Manning et al., 2019). Moreover, the occurrence of CDHW events shows seasonal patterns, with the occurrences of droughts and heatwaves being negatively dependent in winter, but positively dependent in the other three seasons, which could be explained by atmospheric and land-atmosphere interactions. Fourth, a validation experiment based on this positive dependence in summer demonstrated the robustness of the proposed method compared to commonly used methods. We used one station to demonstrate the method, and further studies are needed to validate whether the proposed identification method works well in different climatic zones.

By upscaling the temporal resolution to a finer one, our daily-scale identification captures variations that monthly scales often miss, providing more precise event start and end dates for droughts and CDHW events. This more precise identification could enhance the capacity for detection, assessment, monitoring, and early warning of both drought events and CDHW events.

Furthermore, our definition in relative terms allows for identifying heatwaves and CDHW events across all four seasons, including non-summer periods. This expanded understanding is crucial as it sheds light on the ecological repercussions that extend beyond the confines of the traditional summer-focused perspective. The ecological impacts of CDHW events in non-summer seasons are also significant. For instance, in regions characterized by temperate continental and temperate monsoon climates, CDHW events in non-summer seasons link to wildfire weather (Tian et al., 2011). In such regions, the winter season itself often represents the dry season, characterized by reduced precipitation and frequent strong winds. The dry season becomes even drier when drought conditions co-occur with abnormally high temperatures. This exacerbates the dryness of the soil and the atmosphere, accelerating the drying of forest litter and setting the stage for an elevated risk of wildfires.

*Code availability.* The code of this study is available on request from the corresponding author.

*Author contributions.* B.S.: Investigation, Conceptualization, Methodology, Validation, Formal analysis, Data curation, Writing – original draft. B.D.B and N.V.: Conceptualization, Methodology, Writing – reviewing & editing.

*Competing interests.* The authors declare that they have no known competing financial interests or personal relationships that could have appeared to influence the work reported in this paper.

*Acknowledgements.* The support provided by the China Scholarship Council (CSC) of Baoying Shan at Ghent University is acknowledged. We are grateful to the Royal Meteorological Institute of Belgium for allowing the use of the 120-year Uccle data set. We acknowledge the anonymous reviewers for their constructive feedback and suggestions.

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
