# Peer review of "Identification of compound drought and heatwave events on a daily scale and across four seasons"

_EGUsphere, 2023_

## Author Comment (AC2)

**Response letter to RC2**

In this response letter, we put the text of the reviewers in bold, and the questions are numbered as Q1, Q2 etc. Our answer starts with Answer and is in normal font-weight; The text in the manuscript is put in quotation marks. An empty line separates the questions.

**This study presents a method to identify compound drought/heatwave daily events as well as develop a set of indices that can be used to characterize such events. The paper is well organized and the topic is appropriate for the journal. However, there are a couple of issues that should be addressed before publication (along with some other minor issues outlined below).**

Answer: We sincerely appreciate the reviewer's constructive feedback and thoughtful suggestions concerning the manuscript. Below, we address the issues raised and provide our responses to each of them.

**Q1: In particular, given the definition of drought and heatwave events I'm not sure that a "precise identification of the start and end dates of an event" is really realistic, I would probably focus more on things like duration etc. which are important and somewhat remove the aspect of a specific start/end date.**

Answer: Thank you for raising this important point. You're absolutely correct that achieving an "accurate" identification of drought and heatwave events can be challenging, particularly due to the variability in literature in how these events are defined.

In our study, when we refer to "more precise identification", we are emphasizing the enhancement of the temporal resolution in event identification, specifically at a daily scale. This finer temporal scale allows us to capture variations that might be missed when using monthly scales.

To illustrate this, let's consider an example as discussed in Section 3.2 in the manuscript. In the summer of 1976 at Uccle, the meteorological drought event ended on July 11, followed by heavy precipitation on July 12, which terminated the previous meteorological drought event. If we were to use a monthly scale for identification, this drought event would be recorded as ending on June 30. There is an 11-day difference in the ending dates, and this distinction is particularly significant because it coincided with a heatwave event. Consequently, the corresponding compound events of type d-and-h lasted *18 days* when identified on a daily basis, as opposed to *8 days* when identified on a monthly basis. By employing identification on a daily basis, we can pinpoint starting or ending dates that fall in the middle of the month with more precision, which contributes to a more accurate duration for compound events.

**Q2 & Q3: Also, there is not much discussion on validating the identification of these compound events and although it is a difficult task I would have liked to see some additional text/analysis. A discussion section is also missing that would add some more details on how the results compare in the context of other studies;**

Answer: We acknowledge the importance of discussion and validation in scientific research. Indeed, validating the identification of droughts, heatwaves, and compound events is challenging due to the inherent difficulty in measuring the exact start and end dates. Inspired by the reviewer's question, we have conceived an idea for indirect validation.

We know from well-established physical mechanisms that there exists a strong positive land-atmospheric feedback between droughts and heatwaves during the summer. So, droughts and heatwaves are highly dependent in the summer.

To quantify this dependence, we employ the likelihood multiplication factor (Ridder et al., 2020), denoted as LMF. It is calculated as $\text{LMF} = \frac{P_{\text{d-and-h}}}{P_{\text{d}} \times P_{\text{h}}}$, where $P_{\text{d}}$, $P_{\text{h}}$, and $P_{\text{d-and-h}}$ represent the probability of a day being in drought, heatwave, and CDHW events, respectively. In the case of independence, LMF equals 1; if droughts and heatwaves are positively dependent, LMF exceeds 1 and increases with the strength of dependence.

Back to our identification method, it comprises three crucial steps: pre-identification, removal, and merging. These steps categorize spells into two distinct groups:

A: Drought and heatwave events using the proposed identification method.

B: Short dry and heatwave spells that are removed in the removal step but still fall below (or above) the pre-identification threshold.

So, combined with the known physical mechanisms, if our proposed method is effective, the removed spells should not represent drought or heatwave events. Therefore, we expect that the removed spells (group B) will exhibit less dependence than the events (group A) during the summer. We expect the following behaviors during the summer season:

**(1) LMF$_\text{A}$ should be greater than 1;**

**(2) LMF$_\text{A}$ should be greater than LMF$_\text{B}$.**

We calculated LMF$_\text{A}$, LMF$_\text{B}$, and LMF$_\text{A}$ − LMF$_\text{B}$ in June, July and August. Because different accumulation period lengths and/or pre-identification thresholds for droughts and heatwaves could result in different CDHW events, we consider all possible combinations: $N_{\text{d}} = 15, 30, 45, 60, 90$ d, $N_{\text{h}} = 3, 5, 7, 10, 15$ d, $\text{SPI}_{\text{d}} = $ -0.5, -1, -1.3, $\text{SHI}_{\text{h}} = 0.5, 1, 1.3$, resulting in $5 \times 5 \times 3 \times 3 = 225$ scenarios in total, see Table **??** in the manuscript for the corresponding removal and merging thresholds for each case.

The validation results are presented in Figure 1, and we observed the expected behaviors.

Firstly, LMF$_\text{A}$ consistently exceeds 1 across all scenarios, indicating that the identified droughts and heatwaves exhibit positive dependence during the summer season. LMFB tends to be closer to 1 compared to LMFA, suggesting that the removed dry and heatwave spells are closer to being independent.

Secondly, LMF$_\text{A}$ − LMF$_\text{B}$ >0 in all 225 scenarios, which means the identified droughts and heatwaves always have a stronger positive dependence than the removed dry and heatwave spells.

In existing studies on the identification of drought and heatwave events, it is common practice to employ subjective and fixed minimum duration for excluding minor spells. For instance, many studies retain drought events only if they persist for more than 30 days to focus on significant events (Brunner and Stahl, 2023; Xu et al., 2023; Christian et al., 2021). Similarly, heatwave events are commonly identified as lasting at least 3 consecutive days (Ridder et al., 2020; Yin et al., 2023).

[Figure]

**Figure 1.** $\mathrm{LMF_A}$, $\mathrm{LMF_B}$, and $\mathrm{LMF_A} - \mathrm{LMF_B}$ by using the proposed removal and merging method. $\mathrm{LMF} = \frac{P_{\text{d-and-h}}}{P_{\text{d}} \times P_{\text{h}}}$ and $\mathrm{LMF} = 1$ means independence. A: Drought and heatwave events using the proposed identification method; B: Short dry and heatwave spells that are removed in the removal step but still fall below (or above) the pre-identification threshold.

However, we raise a concern regarding the rationality of applying fixed removal thresholds across varying lengths of accumulation periods and pre-identification thresholds such as in Xu et al. (2023). To investigate this concern, we conducted a similar analysis but using fixed removal thresholds (30 days for droughts and 3 days for heatwaves) instead of the thresholds obtained by the proposed method. In this analysis, we also categorized the identified events into two groups:

70    A2: drought and heatwave events identified using the fixed removal threshold method.

B2: dry and heatwave spells with a duration shorter than the fixed removal thresholds.

Similarly, we calculate $\mathrm{LMF_{A2}}$, $\mathrm{LMF_{B2}}$, and $\mathrm{LMF_{A2}} - \mathrm{LMF_{B2}}$ in all 225 scenarios, as shown in Figure 2. Our aim is to evaluate whether these scenarios exhibited the two expected behaviors.

The results confirmed the first expected behavior: $\mathrm{LMF_{A2}}$ consistently exceeded 1 in all scenarios. However, it is noteworthy that the second expected behavior, $\mathrm{LMF_{A2}} - \mathrm{LMF_{B2}} > 0$, was not always observed. Approximately 16% of cases yielded negative values for $\mathrm{LMF_{A2}} - \mathrm{LMF_{B2}}$, implying that the removed drought and heatwave spells exhibited a larger degree of dependence than the identified events. This outcome deviates from the expected behavior.

[Figure]

**Figure 2.** $LMF_{A2}$, $LMF_{B2}$, and $LMF_{A2} - LMF_{B2}$ by using the subjective minimum duration method. A2: drought and heatwave events identified using the fixed removal threshold method; B2: dry and heatwave spells with a duration shorter than the fixed removal thresholds.

In conclusion, the application of a fixed removal threshold method for various accumulation periods and pre-identification thresholds of droughts and heatwaves introduces the potential risk of generating inconsistent and unreasonable results. In contrast, our removal and merging method effectively addresses this challenge during validation.

In response to the reviewer's feedback, we intend to introduce an additional section in our manuscript that discusses the above validation idea.

**Q4: also, the implications of identifying the compound events can be expanded upon slightly (as the authors seem to attempt in the final sentence of the Conclusions section).**

Answer: We will delve deeper into the implications arising from the identification of CDHW events through providing additional context in the discussion section regarding the ecological impacts of CDHW events occurring across all four seasons.

"In this study, a notable departure from existing research is the recognition that CDHW events are not exclusive to the summer season or extensions thereof. We have observed CDHW events across all seasons, including non-summer periods. This expanded understanding is crucial as it sheds light on the ecological repercussions that extend beyond the confines of the traditional summer-focused perspective. The ecological impacts of CDHW events in non-summer seasons are also significant. For instance, in regions characterized by temperate continental and temperate monsoon climates, CDHW events in non-summer

seasons link to wildfire weather (Tian et al., 2011). In such regions, the winter season itself often represents the dry season, characterized by reduced precipitation and frequent strong winds. When drought conditions co-occur with abnormally high temperatures, the dry season becomes even drier. This exacerbates the aridity of the soil and the atmosphere, accelerating the drying of forest litter. Consequently, the moisture content in the ecosystem is further reduced, setting the stage for an elevated risk of forest fires."

**Q5: Line 29: is the "coarse resolution" referring to spatial, temporal or both? From the next sentence it seems it's temporal but it'd be good to be explicit here.**

Answer: We mean the temporal resolution. We will promptly make the necessary adjustments in the manuscript.

**Q6: Introduction is otherwise well written.**

Answer: Thank you for the reviewer's positive feedback on the introduction of our manuscript.

**Q7: Line 76-80: I don't find this adds much value. Instead, an overview of the experimental design and its rationale would work better here.**

Answer: We will rewrite Lines 76-80 to provide a concise summary of our method and link it with the questions we aim to answer. Here is the revised text:

"This section mainly describes the identification method of CDHW events, which includes three steps. First, we define a drought and heatwave and calculate the indices for addressing research questions (a) and (b). Then, we identify drought and heatwave events from daily indices for addressing question (c). Finally, we identify CDHW events from drought and heatwave events for addressing question (d)".

**Q8: Line 82: why was only one meteorological station used in the study?**

Answer: The decision to use only one station stems from the availability of an extensive dataset spanning 120 years of daily observations at Uccle. This longest-time series in Belgium provides a robust foundation for developing and validating the method proposed in our research. By concentrating on a single station with such a lengthy record, we aim to capitalize on the wealth of historical climate data, allowing us to build and refine our identification method effectively.

However, we recognize the importance of applying this method to more locations to test further its performance, which is one of the main works we are going to do next.

**Q9: Line 84: perhaps use "have been used in many studies" instead of "have been subjected to many studies"?**

Answer: We agree that the phrase "have been used in many studies" is more precise and effectively conveys what we intended. In the upcoming revision, we will make the change to "These high-quality observations have been used in many studies".

**Q10: Line 132-134: to what degree does the choice of distribution change the calculation of the indices? The use of AIC is an objective way of selecting the distribution, but what is the sensitivity of SPI/SHI?**

130 Answer: To address this, we analyzed the frequency of distribution selections based on the lowest AIC values in the calculation of SPI and SHI, as illustrated in Figure 3. In the calculation of SPI, the Gamma distribution was the most frequently selected, followed by the GEV and Burr distributions. For SHI, a wider array of distributions, including the Normal, Gamma, GEV, Inverse Gaussian, EV, Logistic, Loglogistic, and Burr distributions, were observed.

Regarding the sensitivity of SPI and SHI to the choice of distribution, we conducted a comparison between values calculated 135 using fixed distributions (commonly chosen: Gamma and GEV for SPI, Normal and GEV for SHI) and the values in our study. This comparison is depicted in Figure 4. Our findings indicate that the choice of distribution can significantly influence the indices' values, especially in the tails of the distribution that are particularly relevant for extreme events. For example, we observed several instances of problematic SHI values when using the GEV distribution, as indicated by the grey circle. Existing studies, like by Mishra and Singh (2010), Laimighofer and Laaha (2022), also highlighted this uncertainty in indices caused 140 by the choice of the distribution to be fitted.

In response to the reviewer's feedback, we intend to add one paragraph to introduce the choice of distribution for SPI and SHI at the start of the results section and extra text to discuss it in the discussion section.

[Figure]

**Figure 3.** The frequency of the distribution selected with the lowest value for the AIC in the calculation of SPI and SHI

**Q11: Line 148: the etymology of the term "deficiency" refers to quality, I'm wondering if there could be a better term** 145 **to describe "how close two neighboring spells are" (unless I misunderstood the definition). Perhaps "proximity"?**

Answer: Upon reflection, we concur that the term "proximity" is a more suitable choice to convey the concept of closeness or nearness between two spells. We will make the necessary adjustments in our manuscript to replace "deficiency" with "proximity" to enhance clarity and precision in our descriptions.

150 **Q12: Line 168: I would remove the word "Concretely" here, perhaps "Particularly"?**

[Figure]

[Figure]

**Figure 4.** The differences in SPI (left) and SHI (right) values from the choice of the distribution to be fitted.

Answer: We appreciate the reviewer's suggestion regarding using the word "concretely" in the context of our manuscript. However, upon careful consideration of the context and the text flow, we have opted to retain the term "concretely."

After describing the statistical assumption in the last paragraph, the word "concretely" emphasizes the specificity and precision of the actions being taken. It signals to the reader that we are delineating the criteria for removing and merging spells.

**Q13: Line 230: not sure you need a separate subsection for this.**

Answer: We appreciate the reviewer's feedback on the organization of our manuscript. In light of the reviewer's comments, particularly Q2 and Q3, we have recognized the need to include more comprehensive discussions about validating the identification method. In the validation discussion, we will use the probabilities of occurrence, so this content reappears throughout the paper, not only in the results section (Section 3.4: Seasonality) but also in the discussion. Given the significance of this topic, we have opted to create a separate subsection. This decision aims to assist readers in quickly getting the calculations and concepts related to the probabilities of occurrence.

**Q14: Line 309: why specifically these values for Nd, Nh?**

Answer: The reviewer's observation is entirely valid; indeed, there are 225 potential combinations for analyzing compound drought-heatwave (CDHW) events, given the ranges of parameters available ($N_{\mathrm{d}} = 15, 30, 45, 60, 90$ d; $N_{\mathrm{h}} = 3, 5, 7, 10, 15$ d; $\mathrm{SPI_d} = $ -0.5, -1, -1.3; $\mathrm{SHI_h} = $ 0.5, 1, 1.3).

The specific values of $N_{\mathrm{d}} = 15$ d and $N_{\mathrm{h}} = 3$ d were chosen based on their prevalence in the existing literature. These values have been commonly employed in previous studies for the following reasons:

- $N_{\mathrm{d}} = 15$ d: This duration is frequently used in meteorological drought event identification. It strikes a balance between capturing the onset and progression of droughts while avoiding overemphasis on short-lived events.

155

160

165

170

- $N_{\mathrm{h}} = 3$ d: In the case of heatwave events, the Standardized Heatwave Index (SHI) application is still relatively nascent as non-summer heatwaves are not commonly recognized. However, the study by Raei et al. (2018) recommends an accumulation period of $N_{\mathrm{h}} = 3$ d for SHI. This choice aligns with their findings and reflects the current state of research in this area.

By selecting these values, we aimed to maintain consistency with established practices in the field while also ensuring the feasibility of our analysis. We appreciate the reviewer's interest in this aspect of our method and hope this explanation clarifies our rationale for the specific choices.

**Q15: Line 360-374: the Conclusions section is somewhat poorly written. For instance, the last sentence seems out of place while the statement "The results confirmed the effectiveness of the proposed method" needs a bit more context.**

Answer: We appreciate the reviewer's insights and agree that clarity and coherence are vital in summarizing our study's findings. According to this feedback, we rewrite the Conclusion as below:

"We proposed a method for identifying drought, heatwave, and compound events. The identification on a daily scale systematically and objectively removed minor spells and merged mutually dependent ones. The analysis conducted at Uccle demonstrates the effectiveness of the proposed method in several ways. First, the values of removal thresholds exhibit the desired behavior, adapting effectively to varying accumulation periods and pre-identification thresholds. Moreover, the frequency of occurrence of heatwaves and CDHW events has increased in the period 1961–2020 compared to 1901–1960, and the increasing temperatures contribute to the increase in CDHW events, which aligns with the current literature. Besides that, the occurrence of CDHW events shows seasonal patterns, with the occurrences of droughts and heatwaves being negatively dependent in winter but positively dependent in the other three seasons, which physical mechanisms could explain.

As expected, our daily-scale identification captures variations that monthly or weekly scales often miss, providing more precise event start and end dates. This precise identification could enhance the capacity for detection, assessment, monitoring, and early warning of both drought events and CDHW events. Furthermore, our definition in relative terms allows for identifying heatwaves and CDHW events across all four seasons, a feature valuable for ecological research."

**References**

Brunner, M. I. and Stahl, K.: Temporal hydrological drought clustering varies with climate and land–surface processes, Environmental Research Letters, 18, 034 011, 2023.

Christian, J. I., Basara, J. B., Hunt, E. D., Otkin, J. A., Furtado, J. C., Mishra, V., Xiao, X., and Randall, R. M.: Global distribution, trends, and drivers of flash drought occurrence, Nature communications, 12, 6330, 2021.

Laimighofer, J. and Laaha, G.: How standard are standardized drought indices? Uncertainty components for the SPI & SPEI case, Journal of Hydrology, p. 128385, https://doi.org/https://doi.org/10.1016/j.jhydrol.2022.128385, 2022.

Mishra, A. K. and Singh, V. P.: A review of drought concepts, Journal of Hydrology, 391, 202–216, 2010.

Raei, E., Nikoo, M. R., AghaKouchak, A., Mazdiyasni, O., and Sadegh, M.: GHWR, a multi-method global heatwave and warm-spell record and toolbox, Scientific data, 5, 1–15, 2018.

Ridder, N. N., Pitman, A. J., Westra, S., Ukkola, A., Do, H. X., Bador, M., Hirsch, A. L., Evans, J. P., Di Luca, A., and Zscheischler, J.: Global hotspots for the occurrence of compound events, Nature communications, 11, 1–10, 2020.

Tian, X., McRae, D. J., Jin, J., Shu, L., Zhao, F., and Wang, M.: Wildfires and the Canadian Forest Fire Weather Index system for the Daxing'anling region of China, International Journal of Wildland Fire, 20, 963–973, 2011.

Xu, Z., Wu, Z., Shao, Q., He, H., and Guo, X.: From meteorological to agricultural drought: Propagation time and probabilistic linkages, Journal of Hydrology: Regional Studies, 46, 101 329, 2023.

Yin, J., Gentine, P., Slater, L., Gu, L., Pokhrel, Y., Hanasaki, N., Guo, S., Xiong, L., and Schlenker, W.: Future socio–ecosystem productivity threatened by compound drought–heatwave events, Nature Sustainability, 6, 259–272, 2023.

---

## Author Response (AR1)

**Response letter**

In this letter, we give a point-by-point response to the reviews, including all relevant changes made in the manuscript. We put the text of the reviewers in bold, and the questions are numbered as Q1, Q2, etc. Our answer starts with Answer and is in normal font-weight; The text in the manuscript is put in quotation marks. An empty line separates the questions.

**1   Response to RC1**

**This paper discusses ways to determine heatwaves, droughts, and combination of them in a systematic way and at daily scale allowing to better capture the start and end date of the events. They use 120 years of data over Uccle (point) and suggested an objective method to remove minor spells and merge mutually dependent ones and discussed four ways to define a CDHW event. They eventually used the outcomes to study the seasonality of the drought, heatwaves, and combination of them. They found that seasonal patterns, with the occurrences of droughts and heatwaves being negatively dependent in winter but positively dependent in the other three seasons at Uccle.**

**I believe the paper is well structured and well written in almost all aspects: the science within the scope of the journal, relatively novel in the method and concepts, important conclusion, good literature review, and good discussion of the outcomes and associated interpretation. The conclusions are backed up with proper studies, the title and abstract are fine. I only have few minor points:**

Answer: We are truly grateful for the reviewer's positive feedback and thoughtful evaluation of our manuscript. Your comments are highly valuable to us. Here, we address the points raised and provide our responses to each of them.

**Q1: The study is performed using a point measurement that has 120 years of daily temperature and precipitation data. Yet, the number of events, especially compound events, are not much. Is there any recommendation for the minimum length of the data record if someone wants to apply the method over other regions?**

Answer: In the proposed method, three steps are based on fitting probability distributions. Because of this, the longer the time series, the better the fitting: (1) calculating SPI and SHI, (2) events' severity assumed to follow a GEV distribution, and (3) the events' arrivals assumed to follow a Poisson process.

Generally, 30 is widely suggested and used as the minimum length or number for fitting probability distributions in the field of hydrology. Based on this, step (1) asks for at least 30 years of observation. Steps (2) and (3) ask for the final number of identified events to be larger than 30. The tricky part is that the final number of identified events number also depends on the accumulation period and merging threshold besides the observation length.

If we are requested to give a minimum length for the data record, at least 30 years of continuous precipitation/temperature data are needed, but longer records would be preferable. And at the same time, it is important to check the goodness-of-fit performance in steps (2) and (3).

**Q2: Line 26: remove "more" before regionally**

Answer: We sincerely appreciate the reviewer's meticulous review of our manuscript. The suggestion to remove the word 'more' before 'regionally' on Line 26 is duly noted. We will promptly make the adjustments in the manuscript.

**Q3: Line 39: what is (7d)? Fig 7d in their paper?**

Answer: We apologize for the unclear expression: (7 d) in Line 3 means 7 days. We will change the "(7 d)" to "(a week)" to avoid any misunderstanding. Here are the revised sentences (to provide the context, we also copied two sentences before Line 39):

"This coarse scale of the drought index and the mismatch of scales between the drought and heatwave indices could entail some bias on the actual start and/or end dates as well as the severity of droughts, further affecting the identification of CDHW events. A recent study by Mukherjee and Mishra (2021) upscaled the resolution to weekly droughts and daily heatwaves. They identified a drought week when the self-calibrated Palmer Drought Severity Index (Wells et al., 2004) falls below the 10th percentile, thus regarding short dry spells (a week) as drought events. Although the mismatch of scales is partially solved this way, too short dry spells might be identified as drought events."

**Q4: Line 98: not sure how you come up with 91 days? Is the 91 day window for both 30yr and entire record analysis? I think it is better to further clarify this.**

Answer: In Line 98, the 91 refers to 91 windows, resulting from the 30-year moving window approach from 1901 to 2020. Considering the possible non-stationarity of climate variables, we use the past 30 years as the historical reference period, then this method generates windows (1901-1930, 1902-1931, . . . , 1991-2020), resulting in 91 windows in total.

We will promptly clarify this in the manuscript, shown below (to provide the context, we also copied three sentences before Line 98):

"Accounting for the fact that people and ecosystems adapt to a changing climate, we use the past 30 years as the historical reference period and average the values on every day during this period. This average is then regarded as the expected normal temperature on this day, instead of the average during the longest climatology (the period of record). This 30-year moving window approach has been suggested to deal with the climate non-stationarity bias (Hoylman et al., 2022). To examine the impact of this approach, we apply the Mann–Kendall (MK) test for the daily mean temperature in the period 1901–2020 (the period of record) and in the 30-year moving windows (1901-1930, 1902-1931,. . . , 1991-2020; 91 windows in total per day) based on the data described in Subsection 2.1."

**Q5: Line 224: shouldn't it be the sum of "positive" SHI?**

Answer: In the context of our calculation of marginal drought and heatwave severity, we understand the point you've raised about the sign of SPI and SHI values within CDHW events. While most SPI values are indeed negative within CDHW events, they may not always be exclusively so. Similarly, SHI values are typically positive within CDHW events, but exceptions may exist.

To ensure consistency and clarity in our calculations, we have modified the text from

"The marginal drought severity is computed as **the sum of the negative SPI values** within the CDHW event, while the marginal heatwave severity is computed as the sum of the SHI values within the CDHW event."

to:

"The marginal drought severity is calculated as **the negative of the sum of the SPI values** within the CDHW event, while the marginal heatwave severity is calculated as the sum of the SHI values within the CDHW event."

**Q6: Line 254: Check if the orders is right. I think it should be 1.3, 1, and 0.5 instead.**

Answer: Thanks for pointing out this error. The right order should be 1.3, 1, and 0.5. Here is the corrected sentence:

"In the case $N_h = 3$, warm spells shorter than 6, 9, and 13 d are removed when $SHI_h$ ranges over 1.3, 1, and 0.5."

**Q7: Explain why no merging is carried out in some cases in Table 1.**

Answer: In the method we proposed, first the removal procedure is carried out, and subsequently the merging procedure. After the minor spells are removed, the inter-arrival time between spells correspondingly becomes larger, and the neighboring spells are less likely to be mutually dependent. One case that could happen is that after removing minor spells, the inter-arrival time already follows an exponential distribution and thus, there is no more need to carry out the merging procedures. So, no merging occurs in some cases in Table 1. We add a short explanation in subsection 3.2, Removal and merging thresholds.

"Another interesting observation in Table 1 is that no merging is carried out in some cases ($M_d = -\infty$ or $M_h = -\infty$). That is because, in the proposed method, the removal procedure is carried out first and, subsequently, the merging procedure. After the minor spells are removed, the inter-arrival time between spells correspondingly becomes larger, and the neighboring spells are less likely to be mutually dependent. So, it could happen that after removing minor spells, the inter-arrival time already follows an exponential distribution. Thus, there is no more need to carry out the merging procedure.s"

**Q8: For $N_d$=60 and SPI =-1.3, what does Md=51 mean? why all other Mds are much less? Are these results stable?**

Answer: For $N_d = 60$ and SPI =-1.3, $M_d = 51$ means that neighboring spells are merged into one longer event if the proximity is less than 51. The proximity is defined as below:

This study introduces two terms: *total deficiency* and *proximity*. The total deficiency of a time interval $[a,b]$ corresponds to the area enclosed between the SPI (resp. SHI) curve and the pre-identification threshold line. The total drought deficiency ($TD_d$) is calculated as:

$$TD_d = \sum_{i=a}^{b}(SPI_{m,i} - SPI_d),$$

**Table 1.** Removal thresholds ($R_d$), merging thresholds($M_d$), $\Delta N$, and the ratio of $\Delta N$ to $N_1$ , for droughts (a) and heatwaves (b).

| (a) | $R_d$ | | | $M_d$ | | | $\Delta N$ | | | $\Delta N /N_1$ | | |
|---|---|---|---|---|---|---|---|---|---|---|---|---|
| $SPI_d$ | -0.5 | -1 | -1.3 | -0.5 | -1 | -1.3 | -0.5 | -1 | -1.3 | -0.5 | -1 | -1.3 |
| $N_d$=15 | 23 | 14 | 10 | 1 | 15 | 11 | 2 | 15 | 5 | 0.015 | 0.111 | 0.035 |
| $N_d$=30 | 42 | 22 | 20 | 8 | 46 | 1 | 9 | 15 | 1 | 0.115 | 0.185 | 0.015 |
| $N_d$=45 | 44 | 31 | 25 | 1 | -Inf | 1 | 5 | 0 | 2 | 0.053 | 0 | 0.039 |
| $N_d$=60 | 58 | 45 | 24 | 0 | -Inf | 51 | 4 | 0 | 7 | 0.058 | 0 | 0.149 |
| $N_d$=90 | 81 | 59 | 32 | 1 | 9 | 17 | 2 | 4 | 6 | 0.038 | 0.133 | 0.176 |

| (b) | $R_h$ | | | $M_h$ | | | $\Delta N$ | | | $\Delta N /N_1$ | | |
|---|---|---|---|---|---|---|---|---|---|---|---|---|
| $SHI_h$ | 0.5 | 1 | 1.3 | 0.5 | 1 | 1.3 | 0.5 | 1 | 1.3 | 0.5 | 1 | 1.3 |
| $N_h$=3 | 13 | 9 | 6 | 0 | 0 | 3 | 6 | 1 | 9 | 0.027 | 0.005 | 0.040 |
| $N_h$=5 | 16 | 10 | 8 | 0 | -Inf | -Inf | 4 | 0 | 0 | 0.019 | 0 | 0.000 |
| $N_h$=7 | 27 | 11 | 10 | 4 | -Inf | 6 | 3 | 0 | 10 | 0.051 | 0 | 0.076 |
| $N_h$=10 | 23 | 13 | 12 | -Inf | -Inf | 0 | 0 | 0 | 2 | 0 | 0 | 0.015 |
| $N_h$=15 | 29 | 17 | 18 | 2 | -Inf | 11 | 9 | 0 | 1 | 0.073 | 0 | 0.014 |

where $SPI_d$ is the pre-identification threshold for droughts. Similarly, the total heat deficiency ($TD_h$) is calculated as:

$$TD_h = \sum_{i=a}^{b}(SHI_h - SHI_{m,i}),$$

where $SHI_h$ is the pre-identification threshold for heatwaves.

To measure how close two neighboring spells are, we define the *proximity* as the total deficiency of the time window between the spells. The proximity provides a comprehensive view of the period between two spells, accounting for both duration and how far it is from being dry (resp. warm). For example, a small proximity value means the interval is short and/or the SPI (resp. SHI) values are near the pre-identification threshold. In that case, the neighboring spells are more likely to be mutually dependent, suggesting to merge the neighboring spells and the interval between them into one longer spell.

**why all other Mds are much less? Are these results stable?**

To answer the above two questions, we did additional calculations to show the effects of the merging procedure. The number of events after removal and merging thresholds is denoted as $N_1$, and the number of events when carrying out the merging procedures is denoted as $N_2$; the difference between $N_2$ and $N_1$ is denoted as $\Delta N$, $\Delta N = N_2 - N_1$. Table 1 shows the removal threshold, merging threshold, $\Delta N$, and the ratio of $\Delta N$ to $N_1$ for droughts (a) and heatwaves (b).

We can see from Table 1 that a larger $M_d$ doesn't always mean a larger $\Delta N$. For example, $M_d$=51 ($N_d$=60 and SPI =-1.3) results in $\Delta N = 7$. $M_d$=15 ($N_d$=15 and SPI =-1) and $M_d$=46 ($N_d$=30 and SPI =-1) both result in $\Delta N = 15$.

We also can see from the tables that $\Delta N/N_1$ is upper bounded, *i.e.* $\Delta N /N_1$ <0.2 in drought identification and <0.1 in heatwave identification. The merging process is more common in drought identification than in heatwave identification, with

115 larger $\Delta N/N_1$. From this, one could see that the results of $M_\mathrm{d}$ are relatively stable.

**2 Response to RC2**

**This study presents a method to identify compound drought/heatwave daily events as well as develop a set of indices that can be used to characterize such events. The paper is well organized and the topic is appropriate for the journal. However, there are a couple of issues that should be addressed before publication (along with some other minor issues outlined below).**

Answer: We sincerely appreciate the reviewer's constructive feedback and thoughtful suggestions concerning the manuscript. Below, we address the issues raised and provide our responses to each of them.

**Q1: In particular, given the definition of drought and heatwave events I'm not sure that a "precise identification of the start and end dates of an event" is really realistic, I would probably focus more on things like duration etc. which are important and somewhat remove the aspect of a specific start/end date.**

Answer: Thank you for raising this important point. You're absolutely correct that achieving an "accurate" identification of drought and heatwave events can be challenging, particularly due to the variability in literature in how these events are defined. In our study, when we refer to "more precise identification", we are emphasizing the enhancement of the temporal resolution in event identification, specifically at a daily scale. This finer temporal scale allows us to capture variations that might be missed when using monthly scales.

To illustrate this, let's consider an example as discussed in Section 3.2 in the manuscript. In the summer of 1976 at Uccle, the meteorological drought event ended on July 11, followed by heavy precipitation on July 12, which terminated the previous meteorological drought event. If we were to use a monthly scale for identification, this drought event would be recorded as ending on June 30. There is an 11-day difference in the ending dates, and this distinction is particularly significant because it coincided with a heatwave event. Consequently, the corresponding compound events of type d-and-h lasted *18 days* when identified on a daily basis, as opposed to *8 days* when identified on a monthly basis. By employing identification on a daily basis, we can pinpoint starting or ending dates that fall in the middle of the month with more precision, which contributes to a more accurate duration for compound events.

**Q2 & Q3: Also, there is not much discussion on validating the identification of these compound events and although it is a difficult task I would have liked to see some additional text/analysis. A discussion section is also missing that would add some more details on how the results compare in the context of other studies;**

Answer: We acknowledge the importance of discussion and validation in scientific research. Indeed, validating the identification of droughts, heatwaves, and compound events is challenging due to the inherent difficulty in measuring the exact start and end dates. Inspired by the reviewer's question, we have conceived an idea for indirect validation.

We know from well-established physical mechanisms that there exists a strong positive land-atmospheric feedback between droughts and heatwaves during the summer. So, droughts and heatwaves are highly dependent in the summer.

To quantify this dependence, we employ the likelihood multiplication factor (Ridder et al., 2020), denoted as LMF. It is calculated as $\text{LMF} = \frac{P_{\text{d-and-h}}}{P_{\text{d}} \times P_{\text{h}}}$, where $P_{\text{d}}$, $P_{\text{h}}$, and $P_{\text{d-and-h}}$ represent the probability of a day being in drought, heatwave, and CDHW events, respectively. In the case of independence, LMF equals 1; if droughts and heatwaves are positively dependent, LMF exceeds 1 and increases with the strength of dependence.

Back to our identification method, it comprises three crucial steps: pre-identification, removal, and merging. These steps categorize spells into two distinct groups:

A: Drought and heatwave events using the proposed identification method.

B: Short dry and heatwave spells that are removed in the removal step but still fall below (or above) the pre-identification threshold.

So, combined with the known physical mechanisms, if our proposed method is effective, the removed spells should not represent drought or heatwave events. Therefore, we expect that the removed spells (group B) will exhibit less dependence than the events (group A) during the summer. We expect the following behaviors during the summer season:

(1) $\text{LMF}_{\text{A}}$ should be greater than 1;

(2) $\text{LMF}_{\text{A}}$ should be greater than $\text{LMF}_{\text{B}}$.

We calculated $\text{LMF}_{\text{A}}$, $\text{LMF}_{\text{B}}$, and $\text{LMF}_{\text{A}} - \text{LMF}_{\text{B}}$ in June, July and August. Because different accumulation period lengths and/or pre-identification thresholds for droughts and heatwaves could result in different CDHW events, we consider all possible combinations: $N_{\text{d}} = 15, 30, 45, 60, 90$ d, $N_{\text{h}} = 3, 5, 7, 10, 15$ d, $\text{SPI}_{\text{d}} = $ -0.5, -1, -1.3, $\text{SHI}_{\text{h}} = 0.5, 1, 1.3$, resulting in $5 \times 5 \times 3 \times 3 = 225$ scenarios in total, see Table 1 in the manuscript for the corresponding removal and merging thresholds for each case.

The validation results are presented in Figure 1, and we observed the expected behaviors.

Firstly, $\text{LMF}_{\text{A}}$ consistently exceeds 1 across all scenarios, indicating that the identified droughts and heatwaves exhibit positive dependence during the summer season. LMFB tends to be closer to 1 compared to LMFA, suggesting that the removed dry and heatwave spells are closer to being independent.

Secondly, $\text{LMF}_{\text{A}} - \text{LMF}_{\text{B}} > 0$ in all 225 scenarios, which means the identified droughts and heatwaves always have a stronger positive dependence than the removed dry and heatwave spells.

In existing studies on the identification of drought and heatwave events, it is common practice to employ subjective and fixed minimum duration for excluding minor spells. For instance, many studies retain drought events only if they persist for more than 30 days to focus on significant events (Brunner and Stahl, 2023; Xu et al., 2023; Christian et al., 2021). Similarly, heatwave events are commonly identified as lasting at least 3 consecutive days (Ridder et al., 2020; Yin et al., 2023).

However, we raise a concern regarding the rationality of applying fixed removal thresholds across varying lengths of accumulation periods and pre-identification thresholds such as in Xu et al. (2023). To investigate this concern, we conducted a similar analysis but using fixed removal thresholds (30 days for droughts and 3 days for heatwaves) instead of the thresholds obtained by the proposed method. In this analysis, we also categorized the identified events into two groups:

[Figure]

**Figure 1.** $\text{LMF}_\text{A}$, $\text{LMF}_\text{B}$, and $\text{LMF}_\text{A} - \text{LMF}_\text{B}$ by using the proposed removal and merging method. $\text{LMF} = \frac{P_{\text{d-and-h}}}{P_\text{d} \times P_\text{h}}$ and $\text{LMF} = 1$ means independence. A: Drought and heatwave events using the proposed identification method; B: Short dry and heatwave spells that are removed in the removal step but still fall below (or above) the pre-identification threshold.

A2: drought and heatwave events identified using the fixed removal threshold method.

B2: dry and heatwave spells with a duration shorter than the fixed removal thresholds.

Similarly, we calculate $\text{LMF}_\text{A2}$, $\text{LMF}_\text{B2}$, and $\text{LMF}_\text{A2} - \text{LMF}_\text{B2}$ in all 225 scenarios, as shown in Figure 2. Our aim is to evaluate whether these scenarios exhibited the two expected behaviors.

The results confirmed the first expected behavior: $\text{LMF}_\text{A2}$ consistently exceeded 1 in all scenarios. However, it is noteworthy that the second expected behavior, $\text{LMF}_\text{A2} - \text{LMF}_\text{B2} > 0$, was not always observed. Approximately 16% of cases yielded negative values for $\text{LMF}_\text{A2} - \text{LMF}_\text{B2}$, implying that the removed drought and heatwave spells exhibited a larger degree of dependence than the identified events. This outcome deviates from the expected behavior.

In conclusion, the application of a fixed removal threshold method for various accumulation periods and pre-identification thresholds of droughts and heatwaves introduces the potential risk of generating inconsistent and unreasonable results. In contrast, our removal and merging method effectively addresses this challenge during validation.

In response to the reviewer's feedback, we introduce an additional subsection 3.6, A small validation experiment, which discusses the above validation idea.

[Figure]

**Figure 2.** $LMF_{A2}$, $LMF_{B2}$, and $LMF_{A2} - LMF_{B2}$ by using the subjective minimum duration method. A2: drought and heatwave events identified using the fixed removal threshold method; B2: dry and heatwave spells with a duration shorter than the fixed removal thresholds.

**Q4: also, the implications of identifying the compound events can be expanded upon slightly (as the authors seem to attempt in the final sentence of the Conclusions section).**

Answer: We will delve deeper into the implications arising from the identification of CDHW events through providing additional context in the conclusion section regarding the ecological impacts of CDHW events occurring across all four seasons.

"Furthermore, our definition in relative terms allows for identifying heatwaves and CDHW events across all four seasons, including non-summer periods. This expanded understanding is crucial as it sheds light on the ecological repercussions that extend beyond the confines of the traditional summer-focused perspective. The ecological impacts of CDHW events in non-summer seasons are also significant. For instance, in regions characterized by temperate continental and temperate monsoon climates, CDHW events in non-summer seasons link to wildfire weather (Tian et al., 2011). In such regions, the winter season itself often represents the dry season, characterized by reduced precipitation and frequent strong winds. The dry season becomes even drier when drought conditions co-occur with abnormally high temperatures. This exacerbates the dryness of the soil and the atmosphere, accelerating the drying of forest litter and setting the stage for an elevated risk of wildfires."

**Q5: Line 29: is the "coarse resolution" referring to spatial, temporal or both? From the next sentence it seems it's temporal but it'd be good to be explicit here.**

Answer: We mean the temporal resolution. We will promptly make the necessary adjustments in the manuscript.

**Q6: Introduction is otherwise well written.**

Answer: Thank you for the reviewer's positive feedback on the introduction of our manuscript.

**Q7: Line 76-80: I don't find this adds much value. Instead, an overview of the experimental design and its rationale would work better here.**

Answer: We will rewrite Lines 76-80 to provide a concise summary of our method and link it with the questions we aim to answer. Here is the revised text:

"This section mainly describes the proposed method for the identification of CDHW events, which includes three steps. First, we define droughts and heatwaves and calculate the indices for addressing research questions (a) and (b). Then, we identify drought and heatwave events from daily indices for addressing research question (c). Finally, we identify CDHW events from drought and heatwave events for addressing research question (d)."

**Q8: Line 82: why was only one meteorological station used in the study?**

Answer: The decision to use only one station stems from the availability of an extensive dataset spanning 120 years of daily observations at Uccle. This longest-time series in Belgium provides a robust foundation for developing and validating the method proposed in our research. By concentrating on a single station with such a lengthy record, we aim to capitalize on the wealth of historical climate data, allowing us to build and refine our identification method effectively.

However, we recognize the importance of applying this method to more locations to test further its performance, which is one of the main works we are going to do next.

**Q9: Line 84: perhaps use "have been used in many studies" instead of "have been subjected to many studies"?**

Answer: We agree that the phrase "have been used in many studies" is more precise and effectively conveys what we intended. In the upcoming revision, we will make the change to "These high-quality observations have been used in many studies".

**Q10: Line 132-134: to what degree does the choice of distribution change the calculation of the indices? The use of AIC is an objective way of selecting the distribution, but what is the sensitivity of SPI/SHI?**

Answer: To address this, we analyzed the frequency of distribution selections based on the lowest AIC values in the calculation of SPI and SHI (30-day accumulation period for SPI and 3-day accumulation period for SHI), as illustrated in Figure 3. In the calculation of SPI, the Gamma distribution was the most frequently selected, followed by the GEV and Burr distributions. For SHI, a wider array of distributions, including the Normal, Gamma, GEV, Inverse Gaussian, EV, Logistic, Loglogistic, and Burr distributions, were observed.

Regarding the sensitivity of SPI and SHI to the choice of distribution, we conducted a comparison between values calculated using fixed distributions (commonly chosen: Gamma and GEV for SPI, Normal and GEV for SHI) and the values in our study. This comparison is depicted in Figure 4. Our findings indicate that the choice of distribution can significantly influence the indices' values, especially in the tails of the distribution that are particularly relevant for extreme events. For example, we observed several instances of problematic SHI values when using the GEV distribution, as indicated by the grey circle. Existing studies, like by Mishra and Singh (2010), Laimighofer and Laaha (2022), also highlighted this uncertainty in indices caused by the choice of the distribution to be fitted.

In response to the reviewer's feedback, we add one subsection in Results, 3.1 SPI and SHI, to introduce and discuss the choice of distribution for SPI and SHI.

[Figure]

[Figure]

**Figure 3.** The frequency of the distribution selected with the lowest value for the AIC in the calculation of SPI and SHI

[Figure]

[Figure]

**Figure 4.** The differences in SPI (left) and SHI (right) values from the choice of the distribution to be fitted.

**Q11: Line 148: the etymology of the term "deficiency" refers to quality, I'm wondering if there could be a better term to describe "how close two neighboring spells are" (unless I misunderstood the definition). Perhaps "proximity"?**

Answer: Upon reflection, we concur that the term "proximity" is a more suitable choice to convey the concept of closeness or nearness between two spells. We make the necessary adjustments in our manuscript to replace "deficiency" with "proximity" to enhance clarity and precision in our descriptions. The proximity is defined as below:

"In this study, we develop a method to obtain objective thresholds for excluding minor spells and merging mutually dependent ones. First, two indicators are selected: the *duration* (d) and the *proximity*. The duration expresses how long a spell lasts, and equals the number of days in a dry ($D_d$) or a warm spell ($D_h$), while the proximity describes how close two neighboring spells are (Fig.1). To define the proximity, we first define the total deficiency. The total deficiency of a time interval $[a,b]$ corresponds to the area enclosed between the SPI (resp. SHI) curve and the pre-identification threshold line. The total drought deficiency ($TD_d$) and total heatwave deficiency ($TD_h$) are calculated as:

$$TD_d = \sum_{i=a}^{b}(SPI_{m,i} - SPI_d) \tag{1}$$

$$TD_h = \sum_{i=a}^{b}(SHI_h - SHI_{m,i}) \tag{2}$$

where $SPI_d$ and $SHI_h$ are the corresponding pre-identification thresholds.

The drought proximity $C_d$ (resp. heatwave proximity $C_h$) is defined as the total drought deficiency $TD_d$ (resp. $TD_h$) of the time window between two neighboring spells."

**Q12: Line 168: I would remove the word "Concretely" here, perhaps "Particularly"?**

Answer: We appreciate the reviewer's suggestion regarding using the word "concretely" in the context of our manuscript. However, upon careful consideration of the context and the text flow, we have opted to retain the term "concretely."

After describing the statistical assumption in the last paragraph, the word "concretely" emphasizes the specificity and precision of the actions being taken. It signals to the reader that we are delineating the criteria for removing and merging spells.

**Q13: Line 230: not sure you need a separate subsection for this.**

Answer: We appreciate the reviewer's feedback on the organization of our manuscript. In light of the reviewer's comments, particularly Q2 and Q3, we have recognized the need to include more comprehensive discussions about validating the identification method. In the validation discussion, we will use the probabilities of occurrence, so this content reappears throughout the paper, not only in the results section (Section 3.4: Seasonality) but also in the discussion. Given the significance of this topic, we have opted to create a separate subsection. This decision aims to assist readers in quickly getting the calculations and concepts related to the probabilities of occurrence.

**Q14: Line 309: why specifically these values for Nd, Nh?**

Answer: The reviewer's observation is entirely valid; indeed, there are 225 potential combinations for analyzing compound drought-heatwave (CDHW) events, given the ranges of parameters available ($N_d = 15, 30, 45, 60, 90$ d; $N_h = 3, 5, 7, 10, 15$ d; $SPI_d = $ -0.5, -1, -1.3; $SHI_h = 0.5, 1, 1.3$).

The specific values of $N_d = 15$ d and $N_h = 3$ d were chosen based on their prevalence in the existing literature. These values have been commonly employed in previous studies for the following reasons:

- $N_d = 15$ d: This duration is frequently used in meteorological drought event identification. It strikes a balance between capturing the onset and progression of droughts while avoiding overemphasis on short-lived events.

- $N_h = 3$ d: In the case of heatwave events, the Standardized Heatwave Index (SHI) application is still relatively nascent as non-summer heatwaves are not commonly recognized. However, the study by Raei et al. (2018) recommends an accumulation period of $N_h = 3$ d for SHI. This choice aligns with their findings and reflects the current state of research in this area.

By selecting these values, we aimed to maintain consistency with established practices in the field while also ensuring the feasibility of our analysis. We appreciate the reviewer's interest in this aspect of our method and hope this explanation clarifies our rationale for the specific choices.

**Q15: Line 360-374: the Conclusions section is somewhat poorly written. For instance, the last sentence seems out of place while the statement "The results confirmed the effectiveness of the proposed method" needs a bit more context.**

Answer: We appreciate the reviewer's insights and agree that clarity and coherence are vital in summarizing our study's findings. According to this feedback, we rewrite the Conclusion as below:

[revised manuscript text omitted]

---

## Referee Report (RR1)

The manuscript titled 'Identification of compound drought and heatwave events on a daily scale and across four seasons' is well-written and discussed. Compound drought and heatwave (CDHW) are highly relevant and critical issues in today's world. I have the following comments on the paper:

Major comments:
- McKee et al., 1993, is a very old reference, now many drought products are based on daily or weekly data, so I think the author's argument that, generally, droughts are identified based on monthly magnitude is not strong enough.
- Regarding the seasonal CDHW, the definition of heat waves and drought needs to be properly adjusted for summer and winter, as they will have different consequences.
- The introduction section needs improvement. Please specify the novelty of the work in the introduction section, as previous studies have already discussed CDHW at daily scale.
- Only one lat/long data record has been analyzed, which needs to be expanded to a bigger area.
- It will be good to compare the impact of all historical CDHW on crop yield and vegetation.

Minor comments:
- Please expand SPI in line 88. Also, give a first-level definition of what is SPI and SHI in words.
- In the moving window, are you taking mean? Then specify that. (line 95).
- It would be helpful if the spatial and temporal extent of the study is defined in section 2.1.

---

## Author Response (AR2)

**Response letter to editor**

We are pleased that the editor found our responses to be adequate. There are no new comments from the editor. The revised manuscript based on the reviewers' comments is uploaded.

---

## Author Response (AR3)

**Response letter**

In this response letter, we put the text of the reviewers in bold, and the questions are numbered as Q1, Q2 etc. Our answer starts with Answer and is in normal font-weight. The text in the manuscript is put in quotation marks. The revised texts are underlined. An empty line separates the questions.

**1 Response to Referee No. 3**

**The manuscript titled 'Identification of compound drought and heatwave events on a daily scale and across four seasons' is well-written and discussed. Compound drought and heatwave (CDHW) are highly relevant and critical issues in today's world. I have the following comments on the paper:**

**Q1: McKee et al., 1993, is a very old reference now, many drought products are based on daily or Weekly data, so I think the author's argument that, generally, droughts are identified based on monthly magnitude is not strong enough.**

Answer: According to the review's suggestions that the argument is not strong enough, we revised the text by adding the latest references and giving more demonstrations. The entire paragraph is as shown below, where the changed texts are underlined:

"The identification of CDHW events lies at the core of temporal trend and frequency analyses. However, the coarse temporal resolution and scale inconsistency of the indices used hamper the reliability of existing studies. Droughts are commonly identified on the basis of monthly magnitude variations in historical climate variables (McKee et al., 1993; Ridder et al., 2020; Salvador et al., 2020), and therefore completely ignore the intramonthly distribution, while heatwaves are usually obtained on a daily scale. For example, the *3-month* Standardized Precipitation Index (SPI) for meteorological droughts and the *daily* temperature index for heatwaves are a popular pair of indices that are used for identifying compound events (see, e.g., Geirinhas et al. (2021)). This coarse scale of the drought index and the mismatch of scales between the drought and heatwave indices could entail some bias on the actual start and/or end dates as well as the severity of droughts, further affecting the identification of CDHW events. Recent studies have upscaled the scale of drought events to a weekly, 5-day, or daily scale (Mukherjee and Mishra, 2021; Wang et al., 2020; Li et al., 2020; Mo and Lettenmaier, 2015; Yuan et al., 2023). For example, Mukherjee and Mishra (2021) identified a drought week when the self-calibrated Palmer Drought Severity Index (Wells et al., 2004) falls below the 10th percentile, thus regarding short dry spells (a week) as drought events; Wang et al. (2020) calculated daily Standardized Precipitation Evapotranspiration Index (SPEI) values and identified a drought event as a period of consecutive days with SPEI below a given threshold (such as -1.0). While these identification methods enhance the temporal resolution, they also introduce a new challenge. They may result in a large number of short dry spells and/or mutually dependent ones. Consequently, the droughts identified might not be extreme or independent. Therefore, additional efforts are needed

to address the issues of coarse resolution and scale inconsistency."

30

**Q2: Regarding the seasonal CDHW, the definition of heat waves and drought needs to be properly adjusted for summer and winter, as they will have different consequences.**

Answer: We agree that the effects of CDHW in summer or winter seasons may be different, and this difference may change with the research objects, such as the effects of CDHW on insects, crops, and even broader ecological systems. However, we
35 do not opt to change the definition of heat waves or droughts across seasons as this may make the analyses very complicated. Using a unified definition makes it easier for researchers to compare effects across seasons, even though CDHW events in different seasons may cause different (ecological) consequences.

**Q3: The introduction section needs improvement. Please specify the novelty of the work in the introduction section,**
40 **as previous studies have already discussed CDHW at daily scale.**

Answer: We appreciate the feedback on emphasizing the novelty of our work in the introduction. While it's true that previous studies, like the one by Ridder et al. (2020), have discussed CDHW events on a daily scale, these studies often combine monthly drought data with daily heatwave analysis. For example, they calculate monthly Standardized Precipitation Index (SPI) values and then extend these monthly values to daily data, which does not capture the day-to-day variability of droughts.
45 In response to reviewer's suggestion, we have updated the third paragraph of the introduction to clearly outline the innovative aspects of our research, and please see the answers to Q1 or the revised manuscript for details.

**Q4: Only one lat/long data record has been analyzed, which needs to be expanded to a bigger area.**

Answer: The decision to use only one station stems from the availability of an extensive dataset spanning 120 years of daily
50 observations at Uccle. This time series, the longest in Belgium, provides a robust foundation for developing and validating the method proposed in our research. By concentrating on a single station with such a lengthy record, we aim to capitalize on the wealth of historical climate data, allowing us to build and refine our identification method effectively.

However, we recognize the importance of applying this method to more locations to further test its performance. In fact, during the review period of this manuscript, we have applied this method to identify the droughts, heatwaves, pluvials, and cold-
55 waves of England in our latest research. A preprint is available online ($https://papers.ssrn.com/sol3/papers.cfm?abstract_id = 4728550$).

**Q5: It will be good to compare the impact of all historical CDHW on crop yield and vegetation.**

Answer: We also believe that it is interesting to explore the impacts of historical CDHW on crop yield or vegetation.
60 However, it is a heavy workload, and many interesting questions can be discussed. Indeed, we are working on another study about how compound extreme events across growth stages impact the final winter wheat yields in France. We applied the identification method proposed in this manuscript, and the early-stage results indicate significant associations between extreme events and crop yields.

This manuscript, however, is primarily focused on the identification method. We provide a detailed introduction to the concept and calculations of the method. The results are analyzed by comparing them to those obtained using monthly scale identification, showcasing a more precise identification of start and end dates. Furthermore, we have conducted a small validation experiment to highlight the advantages of our method, as detailed in subsection 3.6.

Therefore, we think the reviewer's suggestion is interesting, and we are indeed working on a related topic, but incorporating such an analysis into this paper might divert from its main focus.

**Q6: Please expand SPI in line 88. Also, give a first-level definition of what is SPI and SHI in words.**

Answer: The full name of the abbreviation SPI (Standardized Precipitation Index) was introduced when it first appeared in the introduction (Line 32). We added a definition of SPI and SHI in words at the beginning of subsection 2.2 as below:

"We use the daily SPI (McKee et al., 1993) and SHI (Standardized Heatwave Index) (Raei et al., 2018) as indices to identify droughts and heatwaves, respectively. SPI (resp. SHI) is a standardized index to quantify to what extent precipitation (resp. temperature) deviates from the climatological average. SPI (resp. SHI) describes dry (resp. hot) spells according to the probability of occurrence in a reference period."

**Q7: In the moving window, are you taking mean? Then specify that. (line 95).**

Answer: In the moving window, we use the past 30 years as the historical reference period then we calculate the SHI.

**Q8: It would be helpful if the spatial and temporal extent of the study is defined in section 2.1.**

Answer: In section 2.1, we have introduced that the spatial extent is one station and the temporal extent is from 19010 to 2020, and the text is shown below:

"Daily minimum/maximum temperatures from 1901 to 2020 were acquired from the climatological station of the Royal Meteorological Institute (RMI) of Belgium in Uccle ($50°47'55''$ N, $4°21'29''$ E, 100 m a.s.l.), while daily rainfall was obtained from 10-minute precipitation time series recorded at the same site."

**2 Response to Referee No. 4**

**This paper introduces a method for identifying compound drought/heatwave events on a daily basis. In general, the study is well-written and the chosen topic is suitable for the journal. The authors have addressed most of the comments made by previous reviewers, but there are a few remaining issues that need to be resolved before publication, along with some minor concerns mentioned below.**

**In particular, the authors should clarify the specific term for drought. Droughts are generally categorized into two major types including conventional droughts and flash droughts. The conventional droughts are referred to those events that last longer and they are usually assessed on a monthly scale, while flash droughts are those events that develop and intensify rapidly, which have shorter durations (e.g. 20 days). In this study, while the authors proposed a method to**

**detect droughts on a daily basis, they did not mention flash droughts or use specific method to detect droughts with short duration (e.g. 15, or 20 days). The major problem is that authors defined droughts on a daily basis using daily SPI, which I am afraid that it might not be a right approach. The SPI is used in various studies to define droughts on**
100 **monthly scale, but it cannot detect rapid onset and development of droughts in finer temporal scales (e.g. 15 day in this study), as it might lead to false or unreliable drought signals, because flash droughts cause an intense reduction in soil moisture and a short reduction in precipitation without affecting soil or plants might not be a drought. Therefore, to capture these droughts, meteorological drought indices, including evaporation (potential or actual evaporation) should be used in the analysis, such as SSI (Standardized Soil moisture Index), SPEI or SNPI.**

105 Answer: We understand the reviewer's concerns. The concept of drought is complex, and defining droughts remains a challenging issue. Drought is a natural phenomenon and it relates to multiple Earth's spheres, such as the atmosphere (dry atmosphere), lithosphere (dry land surface), subsurface layers (dry deep land), and hydrosphere (dry runoff). Besides the common sense of "water deficit" in drought definitions, the community has not yet reached a consensus on a perfect drought index. Numerous drought indices exist, each addressing different facets of this natural phenomenon, including the Standardized
110 Precipitation Index (SPI), z-score, Standardized Precipitation Evapotranspiration Index (SPEI, which is derived from SPI and incorporates evapotranspiration), and soil moisture (used for identifying flash droughts by Yuan et al. (2023)), SSI, SNPI, TVDI (temperature vegetation dryness index), GRACE Groundwater Drought Index, and so on.

We chose SPI for several reasons: firstly, SPI is a basic and widely used index; secondly, our final aim is to study compound drought and heatwave events, and heatwaves are defined using temperatures. For droughts, we did not consider the temperature-
115 included indices (such as SPEI) to avoid the double consideration of the temperature. This approach aligns with many studies on CDHW events, such as (Ridder et al., 2020; Salvador et al., 2020; Geirinhas et al., 2021).

We also agree that there could be better indices than SPI for capturing agricultural droughts or flash droughts. Higher temperatures increase evapotranspiration, rapidly depleting soil moisture and affecting agricultural systems, even if the precipitation anomaly is not so extreme. It is interesting to investigate the connections and distinctions between CDHW events and flash
120 droughts, as both are linked to water deficits and are influenced by temperature. In such investigations, the method presented in this paper could be applied to the index that is best suited for the considered application.

**Q1: Specific Comments: Page 2, line 30: what about flash droughts? Recent studies focused on the concurrent occurrence of flash droughts and heatwave. This type of droughts usually occur in finer time scales (e.g. 20 days). So, the**
125 **statement 'Generally, droughts are identified . . . ' might not be true. The authors could at least mention flash droughts in this paragraph. Besides, heatwaves can be the driver of flash droughts in some cases (Mo et al. 2015).**

Answer: According to the reviewer's suggestions, we revised the introduction of droughts and added more references. To avoid repetition, the improved paragraph is seen in the answer to Q1 for Referee No. 3 or the revised manuscript.

130 **Q2: Page 4, line 89: please add 'respectively' to the end of this sentence: 'as indices to identify droughts and heatwaves.'**

Answer: We have added 'respectively' to the end of this sentence, the revised sentence is shown as:

"We use the daily SPI (McKee et al., 1993) and SHI (Standardized Heatwave Index) (Raei et al., 2018) as indices to identify droughts and heatwaves, respectively."

135

**Q3: In the caption of Figure S1, please correct 30-years. It should be ''in the past 30-year'' or ''30 years''.**

Answer: Thanks for pointing out this problem, we modified the sentence as below:

"Results indicate it is effective to account for climate non-stationarity by using the past 30 years as the historical reference period instead of the whole period."

140

**Q4: Line 131-136: while the selection of probability function significantly influences the values of an indicator, why did you not use at least one non-parametric function? The most attractive feature of the non-parametric drought frameworks is that it leads to statistically consistent drought indicators based on different variables (Farahmand and Aghakouchak, 2015). Most drought indicators rely on a representative parametric probability distribution function that fits**
145 **the data. However, a parametric distribution function may not fit the data. Non-parametric drought indicators do not assume representative parametric distributions.**

Answer: We acknowledge the merits of non-parametric drought frameworks highlighted in existing literature, such as Farahmand and AghaKouchak (2015). Our preference for parametric distributions over non-parametric ones was based on two key considerations.

150    First, while non-parametric methods simplify calculations by eliminating the need for data fitting, they require a larger dataset to ensure robustness. Given our study's reliance on 30 years of observational data (considering the non-stationarity of temperature data, the past 30 years is used as the historical reference period), we were concerned about the sufficiency of this period for the reliable application of non-parametric methods. Second, our literature review revealed a relatively limited adoption of non-parametric distributions for drought index calculations; the literature still prefers the traditional way, even for
155    recent studies (see, e.g., Ridder et al. (2020); Salvador et al. (2020)). In this way, we followed the more classic way to calculate the SPI and SHI. We give several commonly used distributions in the literature, and the most fitting distribution for the data is chosen by the criterion of minimizing the Akaike Information Criterion (AIC). Overall, we agree that non-parametric distributions are a possible choice, but we chose a conservative but safe parametric distribution in this study.

160    **Q5: In page 6, line 155, please add description for m and i for both SPI and SHI.**

Answer: In line 155, we introduce the "where $SPI_d$ and $SHI_h$ are the corresponding pre-identification thresholds". There is no need of $m$ and $i$ for $SPI_d$ and $SHI_h$, as for all year $m$ and day $i$, we apply the same threshold.

**Q6: While the authors explain the reason for using data from only one station (mainly because of availability of rela-**
165 **tively long P and T records) in response to previous referees, still I'm not sure whether this approach would be practical in other regions or not. Several factors affect the occurrence, severity and characteristics of droughts, heatwaves, and**

**compound extremes. By using just one station, the authors neglect important factors such as background aridity and weather patterns in their study and it still remains unclear how this methodology works in different climatic zones.**

Answer: We recognize the importance of applying this method to more locations to further test its performance. During the review period of this manuscript, we have applied the proposed removal and merging method to identify droughts, heatwaves, pluvials, and coldwaves in England with 71 years of data in our latest research. Now, the preprint is available online ($https://papers.ssrn.com/sol3/papers.cfm?abstract_id = 4728550$). We explored the association between the severity of extreme climate events and changes in butterfly abundance. The findings indicate that extreme climate events can be effectively identified using this method, which means the method works well in England. To highlight this concern, we add a limitation statement in the conclusion:

"We used one station to demonstrate the method, and further studies are needed to validate whether the proposed identification method works well in different climatic zones."

**Q7: While the authors enhanced the conclusion as suggested by referee2, it still needs further improvement and lacks a comprehensive representation of results and assessments.**

**Q8: In line 419: please include references for this statement 'the increasing temperatures contribute to the increase in CDHW events, which aligns with the current literature.'**

**Q9: In line 421: either further explain this statement 'which could be explained by physical mechanisms.' or add some references.**

**Q10: In line 422: I am not sure about this statement 'our daily-scale identification captures variations that monthly or weekly scales often miss', because I did not see any weekly analysis that confirms this assertion.**

Answer: Questions Q7 to Q10 all point to conclusions. According to the comments, we revised the conclusion by adding the corresponding references, improving the precision of expression, and enhancing the comprehensive representation of results and assessments. The improved conclusions are as follows:

[revised manuscript text omitted]